# Unbiased Compression Saves Communication in Distributed Optimization: When and How Much?

**Yutong He**[*]
Peking University
yutonghe@pku.edu.cn

**Xinmeng Huang**[*]
University of Pennsylvania
xinmengh@sas.upenn.edu

**Kun Yuan**
Peking University
kunyuan@pku.edu.cn

## Abstract

Communication compression is a common technique in distributed optimization that can alleviate communication overhead by transmitting compressed gradients and model parameters. However, compression can introduce information distortion, which slows down convergence and incurs more communication rounds to achieve desired solutions. Given the trade-off between lower per-round communication costs and additional rounds of communication, it is unclear whether communication compression reduces the total communication cost.

This paper explores the conditions under which unbiased compression, a widely used form of compression, can reduce the total communication cost, as well as the extent to which it can do so. To this end, we present the first theoretical formulation for characterizing the total communication cost in distributed optimization with unbiased compressors. We demonstrate that unbiased compression alone does not necessarily save the total communication cost, but this outcome can be achieved if the compressors used by all workers are further assumed independent. We establish lower bounds on the communication rounds required by algorithms using independent unbiased compressors to minimize smooth convex functions and show that these lower bounds are tight by refining the analysis for ADIANA. Our results reveal that using independent unbiased compression can reduce the total communication cost by a factor of up to $\Theta(\sqrt{\min\{n, \kappa\}})$ when all local smoothness constants are constrained by a common upper bound, where $n$ is the number of workers and $\kappa$ is the condition number of the functions being minimized. These theoretical findings are supported by experimental results.

## 1 Introduction

Distributed optimization is a widely used technique in large-scale machine learning, where data is distributed across multiple workers and training is carried out through worker communication. However, dealing with a vast number of data samples and model parameters across workers poses a significant challenge in terms of communication overhead, which ultimately limits the scalability of distributed machine learning systems. To tackle this issue, communication compression strategies [3, 8, 52, 55, 49] have emerged, aiming to reduce overhead by enabling efficient yet imprecise message transmission. Instead of transmitting full-size gradients or models, these strategies exchange compressed gradients or model vectors of much smaller sizes in communication.

There are two common approaches to compression: quantization and sparsification. Quantization [3, 20, 39, 52] maps input vectors from a large, potentially infinite, set to a smaller set of discrete values. In contrast, sparsification [57, 55, 50] drops a certain amount of entries to obtain a sparse vector for communication. In literature [3, 27, 21], these compression techniques are often modeled

---

[*]Equal Contribution. Corresponding Author: Kun Yuan. Kun Yuan is also affiliated with National Engineering Labratory for Big Data Analytics and Applications, and AI for Science Institute, Beijing, China.

37th Conference on Neural Information Processing Systems (NeurIPS 2023).

as a random operator $C$, which satisfies the properties of unbiasedness $\mathbb{E}[C(x)] = x$ and $\omega$-bounded variance $\mathbb{E}\|C(x) - x\|^2 \leq \omega\|x\|^2$. Here, $x$ represents the input vector to be compressed, and $\omega$ is a fixed parameter that characterizes the degree of information distortion. Besides, part of the compressors can also be modeled as biased yet contractive operators [19, 48, 49].

While communication compression efficiently reduces the volume of vectors sent by workers, it suffers substantial information distortion. As a result, algorithms utilizing communication compression require additional rounds of communication to converge satisfactorily compared to algorithms without compression. This adverse effect of communication compression has been extensively observed both empirically [57, 25, 8] and theoretically [21, 49]. Since the extra rounds of communication needed to compensate for the information loss may outweigh the saving in the per-round communication cost from compression, this naturally motivates the following fundamental question:

*Q1. Can unbiased compression alone reduce the total communication cost?*

By "unbiased compression alone", we refer to the compression that solely satisfies the assumptions of unbiasedness and $\omega$-bounded variance without any additional advanced properties. To address this open question, we formulate the total communication cost as the product of the per-round communication cost and the number of rounds needed to reach an $\epsilon$-accurate solution to distributed optimization problems. Using this formulation, we demonstrate the decrease in the per-round communication cost from unbiased compression is completely offset by additional rounds of communication. Therefore, we answer Q1 by showing unbiased compression alone *cannot* ensure a lower total communication cost, even with an optimal algorithmic design, see Sec. 3 for more details. This negative conclusion drives us to explore the next fundamental open question:

*Q2. Under what additional conditions and how much can unbiased compression*
*provably save the total communication cost?*

Fortunately, some pioneering works [40, 32, 33] have shed light on this question. They impose *independence* on unbiased compressors, *i.e.*, the compressed vectors $\{C_i(x_i)\}_{i=1}^n$ sent by workers are mutually independent regardless of the inputs $\{x_i\}_{i=1}^n$. This independence assumption enables an "error cancellation" effect, producing a more accurate compressed vector $n^{-1}\sum_{i=1}^n C_i(x_i)$ and hence incurring fewer additional rounds of communication compared to dependent compressors. Consequently, the decrease in the per-round communication cost outweighs the extra communication rounds, reducing the total communication cost.

However, it remains unclear how much the total communication cost can be reduced *at most* by independent unbiased compression and whether we can develop algorithms to achieve this optimal reduction. Addressing this question poses significant challenges as it necessitates a study of the optimal convergence rate for algorithms using independent unbiased compression.

This paper provides the *first* affirmative answer to this question for convex problems by: (i) establishing lower bounds on convergence rates of distributed algorithms employing independent unbiased compression, and (ii) demonstrating the tightness of these lower bounds by revisiting ADIANA [32] and presenting novel and refined convergence rates nearly attaining the lower bounds. Our results reveal that compared to non-compression algorithms, independent unbiased compression can save the total communication cost by up to $\Theta(\sqrt{\min\{n, \kappa\}})$-fold, where $n$ is the number of workers and $\kappa \in [1, +\infty)$ is the function condition number. Figure 1 provides a simple empirical justification. It shows independent compression (ADIANA i.d.) reduces communication costs compared to no compression (Nesterov's Accelerated algorithm), while dependent compression (ADIANA s.d.) does not, which validates our theory.

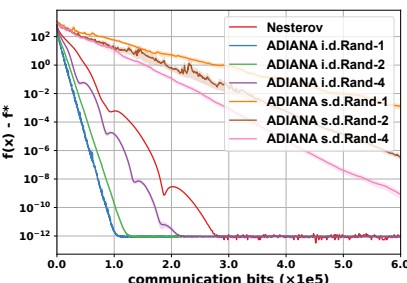

Figure 1: Performance of ADIANA using random-$s$ sparsification compressors with shared (s.d.) or independent (i.d.) randomness against distributed Nesterov's accelerated algorithm with no compression in communication. Experimental descriptions are in Appendix F.1

## 1.1 Contributions

Specifically, our contributions are as follows:

Table 1: Lower and upper bounds on the number of communication rounds for distributed algorithms using unbiased compression to achieve an $\epsilon$-accurate solution. Notations $\Delta, n, L, \mu$ ($\kappa \triangleq L/\mu \geq 1$) are defined in Section 2. $\omega$ is a parameter for unbiased compressors (Assumption 2). $\tilde{\mathcal{O}}$ and $\tilde{\Omega}$ hides logarithmic factors independent of $\epsilon$. GC and SC denote generally-convex and strongly-convex functions respectively.

| Method | GC | SC |
|---|---|---|
| **Lower Bound** | $\tilde{\Omega}\left(\omega\ln(\frac{1}{\epsilon}) + \left(1 + \frac{\omega}{\sqrt{n}}\right)\frac{\sqrt{L\Delta}}{\sqrt{\epsilon}}\right)$ | $\tilde{\Omega}\left(\left(\omega + \left(1 + \frac{\omega}{\sqrt{n}}\right)\sqrt{\kappa}\right)\ln\left(\frac{1}{\epsilon}\right)\right)$ |
| Lower Bound [19]$^\natural$ | $\Omega\left((1+\omega)\frac{\sqrt{L\Delta}}{\sqrt{\epsilon}}\right)$ | $\tilde{\Omega}\left((1+\omega)\sqrt{\kappa}\ln\left(\frac{1}{\epsilon}\right)\right)$ |
| CGD [27]$^\diamond$ | $\mathcal{O}\left((1+\omega)\frac{L\Delta}{\epsilon}\right)$ | $\tilde{\mathcal{O}}\left((1+\omega)\kappa\ln\left(\frac{1}{\epsilon}\right)\right)$ |
| ACGD [32]$^\diamond$ | $\mathcal{O}\left((1+\omega)\frac{\sqrt{L\Delta}}{\sqrt{\epsilon}}\right)$ | $\tilde{\mathcal{O}}\left((1+\omega)\sqrt{\kappa}\ln\left(\frac{1}{\epsilon}\right)\right)$ |
| DIANA [40] | $\mathcal{O}\left(\left(1 + \frac{\omega^2+\omega}{n+\omega}\right)\frac{L\Delta}{\epsilon}\right)$ | $\tilde{\mathcal{O}}\left(\left(\omega + \left(1 + \frac{\omega}{n}\right)\kappa\right)\ln\left(\frac{1}{\epsilon}\right)\right)$ |
| EF21 [49]$^\natural$ | $\tilde{\mathcal{O}}\left((1+\omega)\frac{L\Delta}{\epsilon}\right)$ | $\tilde{\mathcal{O}}\left((1+\omega)\kappa\ln\left(\frac{1}{\epsilon}\right)\right)$ |
| ADIANA [32] | — | $\tilde{\mathcal{O}}\left(\left(\omega + \left(1 + \frac{\omega^{3/4}}{n^{1/4}} + \frac{\omega}{\sqrt{n}}\right)\sqrt{\kappa}\right)\ln\left(\frac{1}{\epsilon}\right)\right)$ |
| CANITA [33]$^\ddagger$ | $\mathcal{O}\left(\omega\frac{\sqrt[3]{L\Delta}}{\sqrt[3]{\epsilon}} + \left(1 + \frac{\omega^{3/4}}{n^{1/4}} + \frac{\omega}{\sqrt{n}}\right)\frac{\sqrt{L\Delta}}{\sqrt{\epsilon}}\right)$ | — |
| NEOLITHIC [19]$^\natural$ | $\tilde{\mathcal{O}}\left((1+\omega)\frac{\sqrt{L\Delta}}{\sqrt{\epsilon}}\right)$ | $\tilde{\mathcal{O}}\left((1+\omega)\sqrt{\kappa}\ln\left(\frac{1}{\epsilon}\right)\right)$ |
| **ADIANA (Thm. 3)** | $\mathcal{O}\left(\omega\frac{\sqrt[3]{L\Delta}}{\sqrt[3]{\epsilon}} + \left(1 + \frac{\omega}{\sqrt{n}}\right)\frac{\sqrt{L\Delta}}{\sqrt{\epsilon}}\right)$ | $\tilde{\mathcal{O}}\left(\left(\omega + \left(1 + \frac{\omega}{\sqrt{n}}\right)\sqrt{\kappa}\right)\ln\left(\frac{1}{\epsilon}\right)\right)$ |

$^\diamond$ Results obtained in the single-worker setting and cannot be extended to the distributed setting.
$^\ddagger$ The rate is obtained by correcting mistakes in the derivations of [33]. See details in Appendix E.
$^\natural$ Results hold without assuming independence across compressors.

- We present a theoretical formalization of the total communication cost in distributed optimization with unbiased compression. With this formulation, we demonstrate that unbiased compression alone is insufficient to save the total communication cost, even with an optimal algorithmic design. This is because any reduction in the per-round communication cost is fully offset by the additional rounds of communication required due to the presence of compression errors.

- We prove lower bounds on the convergence complexity of distributed algorithms using independent unbiased compression to minimize smooth convex functions. Compared to lower bounds when using unbiased compression without independence [19], our lower bounds demonstrate significant improvements when $n$ and $\kappa$ are large, see the first two lines in Table 1. This improvement highlights the importance of independence in unbiased compression.

- We revisit ADIANA [32] by deriving an improved rate for strongly-convex functions and proving a novel convergence result for generally-convex functions. Our rates nearly match the lower bounds, suggesting their tightness and optimality. Our optimal complexities reveal that, compared to non-compression algorithms, independent unbiased compression can decrease total communication costs by up to $\mathcal{O}(\sqrt{\min\{n, \kappa\}})$-fold when all local smoothness constants are constrained by a common upper bound.

- We support our theoretical findings with experiments on both synthetic data and real datasets.

We present the lower bounds, upper bounds, and complexities of state-of-the-art distributed algorithms using independent unbiased compressors in Table 1. With our new and refined analysis, ADIANA nearly matches the lower bounds for both strongly-convex and generally-convex functions.

## 1.2 Related work

**Communication compression.** Two main approaches to compression are extensively explored in literature: quantization and sparsification. Quantization coarsely encodes input vectors into fewer discrete values, *e.g.*, from 32-bit to 8-bit integers [37, 18]. Schemes like Sign-SGD [52, 8] use 1 bit per entry, introducing unbiased random information distortion. Other variants such as Q-SGD [3], TurnGrad [58], and natural compression [20] quantize each entry with more effective bits. In contrast, sparsification either randomly zeros out entries to yield sparse vectors [57], or transmits only the largest model/gradient entries [55].

**Error compensation.** Recent works [52, 59, 55, 4, 49] propose error compensation or feedback to relieve the effects of compression errors. These techniques propagate information loss backward during compression, thus preserving more useful information. Reference [52] uses error compensation for 1-bit quantization, while the work [59] proposes error-compensated quantization for quadratic problems. Error compensation also reduces sparsification-induced errors [55] and is studied for convergence in non-convex scenarios [4]. Recently, the work [49] introduces EF21, an error feedback scheme that compresses only local gradient increments with improved theoretical guarantees.

**Lower bounds.** Lower bounds in optimization set a limit for the performance of a single or a class of algorithms. Prior works have established numerous lower bounds for optimization algorithms [1, 15, 6, 43, 7, 21, 64, 17, 44]. In the field of distributed optimization with communication compression, reference [46] provides an algorithm-specific lower bound for strongly-convex functions, while the work [53] establishes the bit-wise complexities for PL-type problems, which reflect the influence of the number of agents $n$ and dimension $d$, but not the condition number $\kappa$ and the compression $\omega$. In particular, [21] characterizes the optimal convergence rate for all first-order and linear-spanning algorithms, in the stochastic non-convex case, which is later extended by [19] to convex cases.

**Accelerated algorithms with communication compression.** There is a scarcity of academic research on compression algorithms incorporating acceleration, as evident in a limited number of studies [32, 33, 48]. References [32, 33] develop accelerated algorithms with compression in the strongly-convex and generally-convex cases, respectively. For distributed finite-sum problems, accelerated algorithms with compression can further leverage variance-reduction techniques to expedite convergence [48].

**Other communication-efficient strategies.** Other than communication compression studied in this paper, there are a few different techniques to mitigate the communication overhead in distributed systems, including decentralized communication and lazy communication. Notable examples of decentralized algorithms encompass decentralized SGD [12, 34, 30, 64], D2/Exact-Diffusion [56, 62, 61], gradient tracking [47, 60, 29, 2], and their momentum variants [35, 63]. Lazy communication allows each worker to either perform multiple local updates as opposed to a single communication round [38, 54, 41, 24, 14, 22], or by adaptively skipping communication [13, 36].

## 2 Problem setup

This section introduces the problem formulation and assumptions used throughout the paper. We consider the following distributed stochastic optimization problem

$$\min_{x \in \mathbb{R}^d} \quad f(x) = \frac{1}{n} \sum_{i=1}^{n} f_i(x), \tag{1}$$

where the global objective function $f(x)$ is decomposed into $n$ local objective functions $\{f_i(x)\}_{i=1}^{n}$, and each local $f_i(x)$ is maintained by node $i$. Next, we introduce the setup and assumptions.

### 2.1 Function class

We let $\mathcal{F}_{L,\mu}^{\Delta}$ ($0 \leq \mu \leq L$) denote the class of convex and smooth functions satisfying Assumption 1. We define $\kappa \triangleq L/\mu \in [1, +\infty]$ as the condition number of the functions to be optimized. When $\mu > 0$, $\mathcal{F}_{L,\mu}^{\Delta}$ represents strongly-convex functions. Conversely, when $\mu = 0$, $\mathcal{F}_{L,\mu}^{\Delta}$ represents generally-convex functions with $\kappa = \infty$.

**Assumption 1** (CONVEX AND SMOOTH FUNCTION). *We assume each $f_i(x)$ is L-smooth and $\mu$-strongly convex, i.e., there exists constants $L \geq \mu \geq 0$ such that*

$$\frac{\mu}{2}\|y - x\|^2 \leq f_i(y) - f_i(x) - \langle \nabla f_i(x), y - x \rangle \leq \frac{L}{2}\|y - x\|^2$$

*for any $x, y \in \mathbb{R}^d$ and $1 \leq i \leq n$. We further assume $\|x^0 - x^\star\|^2 \leq \Delta$ where $x^\star$ is one of the global minimizers of $f(x) = \frac{1}{n} \sum_{i=1}^{n} f_i(x)$.*

### 2.2 Compressor class

Each worker $i \in \{1, \cdots, n\}$ is equipped with a potentially random compressor $C_i : \mathbb{R}^d \to \mathbb{R}^d$. We let $\mathcal{U}_\omega$ denote the set of all $\omega$-unbiased compressors satisfying Assumption 2, and $\mathcal{U}_\omega^{\mathrm{ind}}$ denote the set of all *independent* $\omega$-unbiased compressors satisfying both Assumption 2 and Assumption 3.

**Assumption 2** (UNBIASED COMPRESSOR). *We assume all compressors $\{C_i\}_{i=1}^n$ satisfy*

$$\mathbb{E}[C_i(x)] = x, \quad \mathbb{E}[\|C_i(x) - x\|^2] \leq \omega \|x\|^2, \quad \forall\, x \in \mathbb{R}^d \tag{2}$$

*for constant $\omega \geq 0$ and any input $x \in \mathbb{R}^d$, where the expectation is taken over the randomness of the compression operator $C_i$.*[2]

**Assumption 3** (INDEPENDENT COMPRESSOR). *We assume all compressors $\{C_i\}_{i=1}^n$ are mutually independent, i.e., outputs $\{C_i(x_i)\}_{i=1}^n$ are mutually independent random variables for any $\{x_i\}_{i=1}^n$.*

## 2.3 Algorithm class

Similar to [19], we consider centralized and synchronous algorithms in which first, every worker is allowed to communicate only directly with a central server but not between one another; second, all iterations/communications are synchronized, meaning that all workers start each of their iterations simultaneously. We further require algorithms to satisfy the so-called "linear-spanning" property, which appears in [9, 10, 21, 19] (see formal definition in Appendix C). Intuitively, this property requires each local model $x_i^k$ to lie in the linear manifold spanned by the local gradients and the received messages at worker $i$. The linear-spanning property is satisfied by all algorithms in Table 1 as well as most first-order methods [42, 28, 23, 65].

Formally, this paper considers a class of algorithms specified by Definition 1.

**Definition 1** (ALGORITHM CLASS). *Given compressors $\{C_i\}_{i=1}^n$, we let $\mathcal{A}_{\{C_i\}_{i=1}^n}$ denote the set of all centralized, synchronous, linear-spanning algorithms admitting compression in which compressor $C_i$, $\forall\, 1 \leq i \leq n$, is applied for the messages sent by worker $i$ to the server.*

For any algorithm $A \in \mathcal{A}_{\{C_i\}_{i=1}^n}$, we define $\hat{x}^k$ and $x_i^k$ as the output of the server and worker $i$ respectively, after $k$ communication rounds.

## 2.4 Convergence complexity

With all the interested classes introduced above, we are ready to define our complexity metric for convergence analysis. Given a set of local functions $\{f_i\}_{i=1}^n \in \mathcal{F}_{L,\mu}^\Delta$, a set of compressors $\{C_i\}_{i=1}^n \in \mathcal{C}$ ($\mathcal{C} = \mathcal{U}_\omega^{\text{ind}}$ or $\mathcal{U}_\omega$), and an algorithm $A \in \mathcal{A}_{\{C_i\}_{i=1}^n}$, we let $\hat{x}_A^t$ denote the output of algorithm $A$ after $t$ communication rounds. The convergence complexity of $A$ solving $f(x) = \frac{1}{n}\sum_{i=1}^n f_i(x)$ under $\{(f_i, C_i)\}_{i=1}^n$ is defined as

$$T_\epsilon(A, \{(f_i, C_i)\}_{i=1}^n) = \min\left\{ t \in \mathbb{N} : \mathbb{E}[f(\hat{x}_A^t)] - \min_x f(x) \leq \epsilon \right\}. \tag{3}$$

This measure corresponds to the number of communication rounds required by algorithm $A$ to achieve an $\epsilon$-accurate optimum of $f(x)$ in expectation.

**Remark 1.** *The measure in (3) is commonly referred to as the communication complexity in literature [51, 21, 31, 32]. However, we refer to it as the convergence complexity here to avoid potential confusion with the notion of "communication complexity" and "total communication cost". This complexity metric has been traditionally used to compare communication rounds used by distributed algorithms [32, 33]. However, it cannot capture the total communication costs of multiple algorithms with different per-round communication costs, e.g., algorithms with or without communication compression. Therefore, it is unable to address the motivating questions Q1 and Q2.*

**Remark 2.** *The definition of $T_\epsilon$ can be independent of the per-round communication cost, which is specified only through the degree of compression $\omega$ (i.e., choice of compressor class). However, to be precise, we may further assume these compressors are non-adaptive with the same fixed per-round communication cost. Namely, the compressors output compressed vectors that can be represented by a fixed and common number of bits. Notably, such hypothesis of non-adaptive cost is widely adopted for practical comparison of communication costs and is valid when input $x$ is bounded or can be encoded with finite bits [3, 59, 20, 22].*

---

[2]Compression is typically employed for input $x$ that is bounded [3, 59] or encoded with finite bits (*e.g.*, float64 numbers) [20]. In these practical scenarios, $\omega$-unbiased compression can be employed with finite bits. For instance, random-$s$ sparsification for $r$-bit $d$-dimensional vectors costs (nearly) $rs = rd/(1+\omega)$ bits per communication where $\omega = d/s - 1$.

# 3 Total communication cost

## 3.1 Fomulation of total communication cost

This section introduces the concept of Total Communication Cost (TCC). TCC can be calculated at both the level of an individual worker and of the overall distributed machine learning system comprising all $n$ workers. In a centralized and synchronized algorithm where each worker communicates compressed vectors of the same dimension, the TCC of the entire system is directly proportional to the TCC of a single worker. Therefore, it is sufficient to use the TCC of a single worker as the metric for comparing different algorithms. In this paper, we let TCC denote the total communication cost incurred by each worker in achieving a desired solution when no ambiguity is present.

Let each worker to be equipped with a non-adaptive compressor with the same fixed per-round communication cost, *i.e.*, the compressor outputs compressed vectors of the same length (size), the TCC of an algorithm $A$ to solve problem (1) using a set of $\omega$-unbiased compressors $\{C_i\}_{i=1}^n$ in achieving an $\epsilon$-accurate optimum can be characterized as

$$\text{TCC}_\epsilon(A, \{(f_i, C_i)\}_{i=1}^n) := \text{per-round cost}(\{C_i\}_{i=1}^n) \times T_\epsilon(A, \{(f_i, C_i)\}_{i=1}^n). \tag{4}$$

## 3.2 A tight lower bound for per-round cost

The per-round communication cost incurred by $\{C_i\}_{i=1}^n$ in (4) will vary with different $\omega$ values. Typically, compressors that induce less information distortion, *i.e.*, associated with a smaller $\omega$, incur higher per-round costs. To illustrate this, we consider random-$s$ sparsification compressors, whose per-round cost corresponds to the transmission of $s$ entries, which depends on parameter $\omega$ through $s = d/(1+\omega)$ (see Example 1 in Appendix A). Specifically, if each entry of the input $x$ is numerically represented with $r$ bits, then the random-$s$ sparsification incurs a per-round cost of $rd/(1+\omega)$ bits up to a logarithm factor.

The following proposition, motivated by the inspiring work [50], establishes a lower bound of TCC when using any compressor satisfying Assumption 2.

**Proposition 1.** *Let $x \in \mathbb{R}^d$ be the input to a compressor $C$ and $b$ be the number of bits needed to compress $x$. Suppose each entry of input $x$ is numerically represented with $r$ bits, i.e., errors smaller than $2^{-r}$ are ignored. Then for any compressor $C$ satisfying Assumption 2, the per-round communciation cost of $C(x)$ is lower bounded by $b = \Omega_r(d/(1+\omega))$ where $r$ is viewed as an absolute number in $\Omega_r(\cdot)$ (See the proof in Appendix B).*

Proposition 1 presents a lower bound on the per-round cost of an arbitrary compressor satisfying Assumption 2. This lower bound is tight since the random-$s$ compressor discussed above can achieve this lower bound up to a logarithm factor. Since $d$ only relates to the problem instance itself and $r$ is often a constant absolute number in practice, *e.g.*, $r = 32$ or $64$, both of which are independent of the choices of compressors and algorithm designs, they can be omitted from the lower bound order. As a result, the TCC in (4) can be lower bounded by

$$\text{TCC}_\epsilon = \Omega((1+\omega)^{-1}) \times T_\epsilon(A, \{(f_i, C_i)\}_{i=1}^n). \tag{5}$$

Notably, when no compression is employed (*i.e.*, $\omega = 0$), $\text{TCC}_\epsilon = \Omega(1) \times T_\epsilon(A, \{(f_i, C_i)\}_{i=1}^n)$ is consistent with the convergence complexity.

# 4 Unbiased compressor alone cannot save total communication cost

With formulation (5), given the number of communication rounds $T_\epsilon$, the total communication cost can be readily characterized. A recent pioneer work [19] characterizes a tight lower bound for $T_\epsilon(A, \{(f_i, C_i)\}_{i=1}^n)$ when each $C_i$ satisfies Assumption 2.

**Lemma 1** ([19], Theorem 1, Informal). *Relying on unbiased compressibility alone, i.e., $\{C_i\}_{i=1}^n \in \mathcal{U}_\omega$, without leveraging additional property of compressors such as mutual independence, the fewest rounds of communication needed by algorithms with compressed communication to achieve an $\epsilon$-accurate solution to distributed strongly-convex and generally-convex optimization problems are lower bounded by $T_\epsilon = \Omega((1+\omega)\sqrt{\kappa}\ln(\mu\Delta/\epsilon))$ and $T_\epsilon = \Omega((1+\omega)\sqrt{L\Delta/\epsilon})$, respectively.*

Substituting Lemma 1 into our TCC lower bound in (5), we obtain $\text{TCC}_\epsilon = \tilde{\Omega}(\sqrt{\kappa}\ln(1/\epsilon))$ or $\Omega(\sqrt{L\Delta/\epsilon})$ in the strongly-convex or generally-convex case, respectively, by relying solely on

unbiased compression. These results do not depend on the compression parameter $\omega$, indicating that *the lower per-round cost is fully compensated by the additional rounds of communication incurred by compressor errors.* Notably, these lower bounds are of the same order as optimal algorithms without compression such as Nesterov's accelerated gradient descent [43, 44], leading to the conclusion:

**Theorem 1.** *When solving convex optimization problems following Assumption 1, any algorithm $A \in \mathcal{A}_{\{C_i\}_{i=1}^n}$ that relies solely on unbiased compression satisfying Assumption 2 cannot reduce the total communication cost compared to not using compression. The best achievable total communication cost with unbiased compression alone is of the same order as without compression.*

Theorem 1 presents a *negative* finding that unbiased compression alone is insufficient to reduce the total communication cost, even with an optimal algorithmic design.[3] Meanwhile, it also implies that to develop algorithms that provably reduce the total communication cost, one must leverage compressor properties beyond $\omega$-unbiasedness as defined in (2). Fortunately, mutual independence is one such property which we discuss in depth in later sections.

## 5 Independent unbiased compressor provably saves communication

### 5.1 An intuition on why independence can help

A series of works [40, 32, 33] have shown theoretical improvements in the total communication cost by imposing independence across compressors, *i.e.*, $\{C_i\}_{i=1}^n \in \mathcal{U}_\omega^{\mathrm{ind}}$. The intuition behind the role of independence among worker compressors can be illustrated by a simple example where workers intend to transmit the same vector $x$ to the server. Each worker $i$ sends a compressed message $C_i(x)$ that adheres to Assumption 2. Consequently, the aggregated vector $n^{-1} \sum_{i=1}^n C_i(x)$ is an unbiased estimate of $x$ with variance

$$\mathbb{E}\left[\left\|\frac{1}{n}\sum_{i=1}^n C_i(x) - x\right\|^2\right] = \frac{1}{n^2}\left(\sum_{i=1}^n \mathbb{E}[\|C_i(x) - x\|^2] + \sum_{i \neq j} \mathbb{E}[\langle C_i(x) - x, C_j(x) - x\rangle]\right) \quad (6)$$

If the compressed vectors $\{C_i(x)\}_{i=1}^n$ are further assumed to be independent, *i.e.*, $\{C_i\}_{i=1}^n \in \mathcal{U}_\omega^{\mathrm{ind}}$, then the cancellation of cross error terms leads to the following equation:

$$\mathbb{E}\left[\left\|\frac{1}{n}\sum_{i=1}^n C_i(x_i) - x\right\|^2\right] = \frac{1}{n^2}\sum_{i=1}^n \mathbb{E}[\|C_i(x) - x\|^2] \leq \frac{\omega}{n}\|x\|^2. \quad (7)$$

We observe that the mutual independence among unbiased compressors leads to a decreased variance, which corresponds to the information distortion, of the aggregated message. Remarkably, this reduction is achieved by a factor of $n$ compared to the transmission of a single compressor. Therefore, the independence among the compressors plays a pivotal role in enhancing the accuracy of the aggregated vector, consequently reducing the number of required communication rounds.

On the contrary, in cases where independence is not assumed and no other properties of compressors can be leveraged, the use of Cauchy's inequality only allows us to bound variance (6) as follows:

$$\mathbb{E}\left[\left\|\frac{1}{n}\sum_{i=1}^n C_i(x) - x\right\|^2\right] \leq \frac{1}{n}\sum_{i=1}^n \mathbb{E}[\|C_i(x) - x\|^2] \leq \omega\|x\|^2. \quad (8)$$

It is important to note that the upper bound $\omega\|x\|^2$ can only be attained when the compressors $\{C_i\}_{i=1}^n$ are identical, indicating that this bound cannot be generally improved further. By comparing (7) and (8), we can observe that the variance of the aggregated vector achieved through unbiased compression with independence can be $n$ times smaller than the variance achieved without independence.

### 5.2 Convergence lower bounds with independent unbiased compressors

While mutual independence can boost the unbiased worker compressors, it remains unclear how much the total communication cost can be reduced *at most* by independent unbiased compression

---

[3]The theoretical results can vary from practical observations due to the particularities of real datasets with which compressed algorithms can enjoy faster convergence, compared to the minimax optimal rates (*e.g.*, ours and [19]) justified without resorting any additional condition.

and how to develop algorithms to achieve this optimal reduction. The following subsections aim to address these open questions.

Following the formulation in (5), to establish the best achievable total communication cost using *independent unbiased compression*, we shall study tight lower bounds on the number of communication rounds $T_\epsilon$ to achieve an $\epsilon$-accurate solution, which is characterized by the following theorem.

**Theorem 2.** *For any $L \geq \mu \geq 0$, $n \geq 2$, the following results hold. See the proof in Appendix C.*

- **Strongly-convex:** *For any $\Delta > 0$, there exists a constant $c_\kappa$ only depends on $\kappa \triangleq L/\mu$, a set of local loss functions $\{f_i\}_{i=1}^n \in \mathcal{F}_{L,\mu>0}^\Delta$, independent unbiased compressors $\{C_i\}_{i=1}^n \in \mathcal{U}_\omega^{\mathrm{ind}}$, such that the output $\hat{x}$ of any $A \in \mathcal{A}_{\{C_i\}_{i=1}^n}$ starting from $x^0$ requires*

$$T_\epsilon(A, \{(f_i, C_i)\}_{i=1}^n) = \Omega\left(\left(\omega + \left(1 + \frac{\omega}{\sqrt{n}}\right)\sqrt{\kappa}\right)\ln\left(\frac{\mu\Delta}{\epsilon}\right)\right)$$

*rounds of communication to reach $\mathbb{E}[f(\hat{x})] - \min_x f(x) \leq \epsilon$ for any $0 < \epsilon \leq c_\kappa \mu\Delta$.*

- **Generally-convex:** *For any $\Delta > 0$, there exists a constant $c = \Theta(1)$, a set of local loss functions $\{f_i\}_{i=1}^n \in \mathcal{F}_{L,0}^\Delta$, independent unbiased compressors $\{C_i\}_{i=1}^n \in \mathcal{U}_\omega^{\mathrm{ind}}$, such that the output $\hat{x}$ of any $A \in \mathcal{A}_{\{C_i\}_{i=1}^n}$ starting from $x^0$ requires at least*

$$T_\epsilon(A, \{(f_i, C_i)\}_{i=1}^n) = \Omega\left(\omega\ln\left(\frac{L\Delta}{\epsilon}\right) + \left(1 + \frac{\omega}{\sqrt{n}}\right)\left(\frac{L\Delta}{\epsilon}\right)^{\frac{1}{2}}\right)$$

*rounds of communication to reach $\mathbb{E}[f(\hat{x})] - \min_x f(x) \leq \epsilon$ for any $0 < \epsilon \leq cL\Delta$.*

**Consistency with prior works.** The lower bounds established in Theorem 2 are consistent with the best-known lower bounds in previous literature. When $\omega = 0$, our result reduces to the lower bound for distributed first-order algorithms established by Y. Nesterov in [43]. When $n = 1$, our result reduces to the lower bound established in [19] for the single-node case.

**Independence improves lower bounds.** A recent work [19] establishes lower bounds for unbiased compression without the independence assumption, listed in the second row of Table 1. Compared to these results, our lower bound in Theorem 2 replaces $\omega$ with $\omega/\sqrt{n}$, showing a reduction in order. This reduction highlights the role of independence in unbiased compression. To better illustrate the reduction, we take the strongly-convex case as an example. The ratio of the number of communication rounds $T_\epsilon$ under unbiased compression with independence to the one without independence is:

$$\frac{\omega + (1 + \omega/\sqrt{n})\sqrt{\kappa}}{(1 + \omega)\sqrt{\kappa}} = \frac{1}{1 + \omega} + \frac{\omega}{1 + \omega}\left(\frac{1}{\sqrt{n}} + \frac{1}{\sqrt{\kappa}}\right) = \Theta\left(\frac{1}{\min\{1 + \omega, \sqrt{n}, \sqrt{\kappa}\}}\right). \quad (9)$$

Clearly, using independent unbiased compression can allow algorithms to converge faster, by up to a factor of $\Theta(\sqrt{\min\{n, \kappa\}})$ (attained at $\omega \gtrsim \sqrt{\min\{n, \kappa\}}$), in terms of the number of communication rounds, compared to the best algorithms with unbiased compressors but without independence.

**Total communication cost.** Substituting Theorem 2 into the TCC formulation in (5), we can obtain the TCC of algorithms using independent unbiased compression. Comparing this with algorithms without compression, such as Nesterov's accelerated algorithm, and using the relations in (9), we can demonstrate that independent unbiased compression can reduce the total communication cost. Such reduction can be up to $\Theta(\sqrt{\min\{n, \kappa\}})$ by using compressors with $\omega \gtrsim \sqrt{\min\{n, \kappa\}}$, e.g., random-$s$ sparsification with $s \lesssim d/\sqrt{\min\{n, \kappa\}}$.

### 5.3 ADIANA: a unified optimal algorithm

By comparing existing algorithms using independent unbiased compression, such as DIANA, ADIANA, and CANITA, to our established lower bounds in Table 1, it becomes clear that there is a noticeable gap between their convergence complexities and our established lower bounds. This gap could indicate that these algorithms are suboptimal, but it could also mean that our lower bounds are loose. As a result, our claim that using independent unbiased compression reduces the total communication cost by up to $\Theta(\sqrt{\min\{n, \kappa\}})$ times is not well-grounded yet. In this section, we address this issue by revisiting ADIANA [32] (Algorithm 1) and providing novel and refined convergence results in both strongly- and generally-convex cases.

**Algorithm 1:** ADIANA

**Input:** Scalars $\{\theta_{1,k}\}_{k=0}^{T-1}, \theta_2, \alpha, \beta, \{\gamma_k\}_{k=0}^{T-1}, \{\eta_k\}_{k=0}^{T-1}, p$.

Initialize $w^0 = x^0 = y^0 = z^0 = h^0 = h_i^0, \forall 1 \le i \le n$.

**for** $k = 0, \cdots, T-1$ **do**

> **On server:**
> Update $x$: $x^k = \theta_{1,k} z^k + \theta_2 w^k + (1 - \theta_{1,k} - \theta_2) y^k$ and broadcast to all workers;
> **On all workers in parallel:**
> Compress the increment of local gradient $m_i^k = C_i(\nabla f_i(x^k) - h_i^k)$ and send to the server;
> Compress the increment of local gradient $c_i^k = C_i(\nabla f_i(w^k) - h_i^k)$ and send to the server;
> Update local shift $h_i^{k+1} = h_i^k + \alpha c_i^k$;
> **On server:**
> Aggregate received compressed message $g^k = h^k + \frac{1}{n} \sum_{i=1}^n m_i^k$;
> Update shift $h^{k+1} = h^k + \alpha \frac{1}{n} \sum_{i=1}^n c_i^k$;
> Apply gradient descent $y^{k+1} = x^k - \eta_k g^k$;
> Update $z$: $z^{k+1} = \beta z^k + (1-\beta)x^k + \frac{\gamma_k}{\eta_k}(y^{k+1} - x^k)$;
> Update $w$: $w^{k+1} = \begin{cases} y^k, & \text{with probability } p, \\ w^k, & \text{with probability } 1-p; \end{cases}$

**Output:** $\hat{x} = w^T$ if $f(w^T) \le f(y^T)$ else $\hat{x} = y^T$.

In the strongly-convex case, we refine the analysis of [32] by: (i) adopting new parameter choices where the initial scalar $\theta_2$ is delicately chosen instead of being fixed as $\theta_2 = 1/2$ in [32], (ii) balancing different terms in the construction of the Lyapunov function. While we do not modify the algorithm design, our technical ingredients are necessary to obtain an improved convergence rate. In the generally-convex case, we provide the *first* convergence result for ADIANA, which is missing in literature to our knowledge. In both strongly- and generally-convex cases, our convergence results (nearly) match the lower bounds in Theorem 2. This verifies the tightness of our lower bounds for both the convergence complexity and the total communication cost. In particular, our results are:

**Theorem 3.** *For any $L \ge \mu \ge 0$, $\Delta \ge 0$, $n \ge 1$, and precision $\epsilon > 0$, the following results hold. See the proof in Appendix D.*

- **Strongly-convex:** *If $\mu > 0$, by setting parameters $\eta_k \equiv \eta = n\theta_2/(120\omega L)$, $\theta_{1,k} \equiv \theta_1 = 1/(3\sqrt{\kappa})$, $\alpha = p = 1/(1+\omega)$, $\gamma_k \equiv \gamma = \eta/(2\theta_1 + \eta\mu)$, $\beta = 2\theta_1/(2\theta_1 + \eta\mu)$, and $\theta_2 = 1/(3\sqrt{n} + 3n/\omega)$, ADIANA requires*

$$\mathcal{O}\left(\left(\omega + \left(1 + \frac{\omega}{\sqrt{n}}\right)\sqrt{\kappa}\right)\ln\left(\frac{L\Delta}{\epsilon}\right)\right)$$

  *rounds of communication to reach $\mathbb{E}[f(\hat{x})] - \min_x f(x) \le \epsilon$.*

- **Generally-convex:** *If $\mu = 0$, by setting parameters $\alpha = 1/(1+\omega)$, $\beta = 1$, $p = \theta_2 = 1/(3(1+\omega))$, $\theta_{1,k} = 9/(k + 27(1+\omega))$, $\gamma_k = \eta_k/(2\theta_{1,k})$, and*

$$\eta_k = \min\left\{\frac{k+1+27(1+\omega)}{9(1+\omega)^2(1+27(1+\omega))L}, \frac{3n}{200\omega(1+\omega)L}, \frac{1}{2L}\right\},$$

  *ADIANA requires*

$$\mathcal{O}\left((1+\omega)\sqrt[3]{\frac{L\Delta}{\epsilon}} + \left(1 + \frac{\omega}{\sqrt{n}}\right)\sqrt{\frac{L\Delta}{\epsilon}}\right)$$

  *rounds of communication to reach $\mathbb{E}[f(\hat{x})] - \min_x f(x) \le \epsilon$.*

**Tightness of our lower bounds.** Comparing the upper bounds in Theorem 3 with the lower bounds in Theorem 2, ADIANA attains the lower bound in the strongly-convex case up to a $\ln(\kappa)$ factor, implying the tightness of our lower bound and ADIANA's optimality. In the generally-convex case, the upper bound matches the lower bound's dominating term $(1 + \omega/\sqrt{n})\sqrt{L\Delta/\epsilon}$ but mismatches the smaller term. This shows the tightness of our lower bound and ADIANA's optimality in the

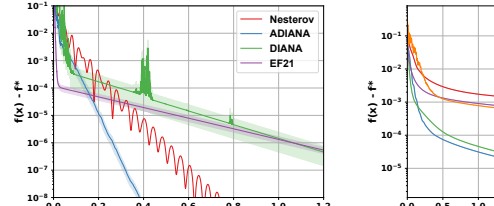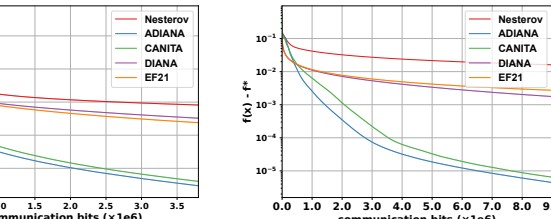

Figure 2: Convergence results of various distributed algorithms on a synthetic least squares problem (left), logistic regression problems with dataset a9a (middle) and w8a (right). The $y$-axis represents $f(\hat{x}) - f^{\star}$ and the $x$-axis indicates the total communicated bits sent by per worker.

high-precision regime $\epsilon < L\Delta \left(\frac{1+\omega/\sqrt{n}}{1+\omega}\right)^6$. Our refined rates for ADIANA are state-of-the-art among existing algorithms using independent unbiased compression.

## 6 Experiments

In this section, we empirically compare ADIANA with DIANA [32], EF21 [49], and CANITA [33] using unbiased compression, as well as Nesterov's accelerated algorithm [43] which is an optimal algorithm when no compression is employed. We conduct experiments on least-square problems (strongly-convex) with synthetic datasets as well as logistic regression problems (generally-convex) with real datasets. In all experiments, we measure the total communicated bits sent by a single worker, which is calculated through *communication rounds to acheive an $\epsilon$-accurate solutions × per-round communicated bits*. All curves are averaged over 20 trials with the region of standard deviations depicted. Due to the space limit, we only provide results with random-$s$ compressors here. More experimental results can be found in Appendix F.2.

**Least squares.** Consider a distributed least-square problem (1) with $f_i(x) := \frac{1}{2}\|A_i x - b_i\|^2$, where $A_i \in \mathbb{R}^{M \times d}$ and $b_i \in \mathbb{R}^M$ are randomly generated. We set $d = 20$, $n = 400$, and $M = 25$, and generate $A_i$'s by randomly generating a Gaussian matrix in $\mathbb{R}^{nM \times d}$, then modify its condition number to $10^4$ through the SVD decomposition, and finally distribute its rows to all $A_i$. We use independent random-1 compressors for communication compression. The results are depicted in Fig. 2 (left) where we observe ADIANA beats all baselines in terms of the total communication cost. We do not compare with CANITA since it does not have theoretical guarantees for strongly-convex problems.

**Logistic regression.** Consider a distributed logistic regression problem (1) with $f_i(x) := \frac{1}{M}\sum_{m=1}^{M} \ln(1 + \exp(-b_{i,m} a_{i,m}^\top x))$, where $\{(a_{i,m}, b_{i,m})\}_{1 \le i \le n, 1 \le m \le M}$ are datapoints in a9a and w8a datasets from LIBSVM [11]. We set $n = 400$ and choose independent random-$\lfloor d/20 \rfloor$ compressors for algorithms with compressed communication. The results are as shown in Fig. 2 (middle and right). Again, we observe that ADIANA outperforms all baselines.

**Influence of independence in unbiased compression.** We also construct a delicate quadratic problem to validate the role of independence in unbiased compression to save communication, see Fig. 1. Experimental details are in Appendix F.1. We observe that ADIANA with independent random-$s$ compressors saves more bits than Nesterov's accelerated algorithm while random-$s$ compressors of shared randomness do not. Furthermore, more aggresive compression, *i.e.*, a larger $\omega$, saves more communication costs in total. These observations are consistent with our theories implied in (9).

## 7 Conclusion

This paper clarifies that unbiased compression alone cannot save communication, but this goal can be achieved by further assuming mutual independence between compressors. We also demonstrate the saving can be up to $\Theta(\sqrt{\min\{n, \kappa\}})$. Future research can explore when and how much biased compressors can save communication in non-convex and stochastic scenarios.

## 8 Acknowledgment

This work is supported by NSFC Grant 12301392, 92370121, and 12288101.

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

## A  Random sparsification

We illustrate the random-$s$ sparsification here. More examples of unbiased compressors can be found in literature [50].

**Example 1** (RANDOM-$s$ SPARSIFICATION). *For any $x \in \mathbb{R}^d$, the random-$s$ sparsification is defined by $C(x) := \frac{d}{s}(\xi \odot x)$ where $\odot$ denotes the entry-wise product and $\xi \in \{0,1\}^d$ is a uniformly random binary vector with $s$ non-zero entries. This random-$s$ sparsification operator $C$ satisfies Assumption 2 with $\omega = d/s - 1$. When each entry of the input $x$ is represented with $r$ bits, random-$s$ sparsification compressor takes $rs$ bits to transmit $s$ entries and $\log_2 \binom{d}{s}$ bits to transmit the indices of $s$ transmitted entries, resulting in a total $\frac{rd}{1+\omega} + \log_2 \binom{d}{s}$ bits in each communication round, see [50, Table 1].*

## B  Proof of Proposition 1

We first recall a result proved by [50].

**Lemma 2** ([50], Theorem 2). *Let $C : \mathbb{R}^d \to \mathbb{R}^d$ be any unbiased compressors satisfying 2 and $b$ be the total number of bits needed to encode the compressed vector $C(x)$ for any $x \in \mathbb{R}^d$. If each entry of the input $x$ is represented with $r$ bits, it holds that $\max\{\frac{\omega}{1+\omega}, 4^{-r}\}4^{b/d} \geq 1$.*

Using Lemma 2, when $\omega/(1+\omega) \leq 4^{-r}$, i.e., $\omega \leq (4^r - 1)^{-1} \leq 1/3$, we have $(1+\omega) = \Theta(1)$ and $b \geq rd = \Omega_r(d/(1+\omega))$, where $r$ is regarded as a constant in $\Omega_r(\cdot)$. When $\omega/(1+\omega) \geq 4^{-r}$, we have

$$b \geq d\log_4(1 + \omega^{-1}) = d\ln(1 + \omega^{-1})/\ln(4) \geq d\frac{\omega^{-1}}{\ln(4)(1 + \omega^{-1})} = \Omega_r\left(\frac{d}{1+\omega}\right),$$

where we use the inequality $\ln(1 + t) \geq t/(1 + t)$ with $t = \omega^{-1} \geq 0$.

## C  Proof of Theorem 2

Following [5, 9], we denote the $k$-th coordinate of a vector $x \in \mathbb{R}^d$ by $[x]_k$ for $k = 1, \ldots, d$, and let $\mathrm{prog}(x)$ be

$$\mathrm{prog}(x) := \begin{cases} 0, & \text{if } x = 0, \\ \max_{1 \leq k \leq d}\{k : [x]_k \neq 0\}, & \text{otherwise.} \end{cases}$$

Similarly, for a set of multiple points $\mathcal{X} = \{x_1, x_2, \ldots\}$, we define $\mathrm{prog}(\mathcal{X}) := \max_{x \in \mathcal{X}} \mathrm{prog}(x)$. We call a function $f$ zero-chain if it satisfies

$$\mathrm{prog}(\nabla f(x)) \leq \mathrm{prog}(x) + 1, \quad \forall x \in \mathbb{R}^d,$$

which implies that starting from $x^0 = 0$, a single gradient evaluation can only earn at most one more non-zero coordinate for the model parameters.

Let us now illustrate the setup of distributed optimization with communication compression. For any $t \geq 1$, we consider the $t$-th communication round, which begins with the server broadcasting a vector denoted as $u^t$ to all workers. We initialize $u^1$ as $x^0$. Upon receiving the vector $u^t$ from the server, each worker performs necessary algorithmic operations, and the round concludes with each worker sending a compressed message back to the server.

We denote $v_i^t$ as the vector that worker $i$ aims to send in the $t$-th communication round before compression, and $\hat{v}_i^t$ as the compressed vector that will be received by the server, i.e., $\hat{v}_i^t = C_i(v_i^t)$. While we require communication to be synchronous among workers, we do not impose restrictions on the number of gradient queries made by each worker within a communication round. We use $\mathcal{Y}_i^t$ to represent the set of vectors at which worker $i$ makes gradient queries in the $t$-th communication round, after receiving $u^t$ but before sending $\hat{v}_i^t$.

Following the above description, we now formally state the linear spanning property in the setting of centralized distributed optimization with communication compression.

**Definition 2** (LINEAR-SPANNING ALGORITHMS). *We say a distributed algorithm $A$ is linear-spanning if, for any $t \geq 1$, the following conditions hold:*

1. *The server can only send a vector in the linear manifold spanned by all the past received messages, sent messages, i.e., $u^t \in \text{span}\left(\{u^r\}_{r=1}^{t-1} \cup \{\hat{v}_i^r : 1 \le i \le n\}_{r=1}^{t-1}\right)$.*

2. *Worker $i$ can only query at vectors in the linear manifold spanned by its past received messages, compressed messages, and gradient queries, i.e., $\mathcal{Y}_i^t \subseteq \text{span}\left(\{u^r\}_{r=1}^t \cup \{\nabla f_i(y) : y \in \mathcal{Y}_i^r\}_{r=1}^{t-1} \cup \{\hat{v}_i^r\}_{r=1}^{t-1}\right)$.*

3. *Worker $i$ can only send a vector in the linear manifold spanned by its past received messages, compressed messages, and local gradient queries, i.e., $v_i^t \in \text{span}\left(\{u^r\}_{r=1}^t \cup \{\nabla f_i(y) : y \in \mathcal{Y}_i^r\}_{r=1}^t \cup \{\hat{v}_i^r\}_{r=1}^{t-1}\right)$.*

4. *After $t$ communication rounds, the server can only output a model in the linear manifold spanned by all the past received messages, sent messages, i.e., $\hat{x}^t \in \text{span}\left(\{u^r\}_{r=1}^t \cup \{\hat{v}_i^r : 1 \le i \le n\}_{r=1}^t\right)$.*

In essence, when starting from $x^0 = 0$, the above linear-spanning property requires that any expansion of non-zero coordinates in vectors held by worker $i$ (e.g., $\mathcal{Y}_i^t$, $v_i^t$) are attributed to its past local gradient updates, local compression, or synchronization with the server. Meanwhile, it also requires that any expansion of non-zero coordinate in vectors held, including the final algorithmic output, in the server is due to the received compressed messages from workers.

Without loss of generality, we assume algorithms to start from $x^0 = 0$ throughout the proofs. When $\{f_i\}_{i=1}^n$ are further assumed to be zero-chain, following Definition 2, one can easily establish by induction that for any $t \ge 1$,

$$\max_{1 \le r \le t} \text{prog}(u^r) \le \max_{1 \le r < t} \max_{1 \le i \le n} \text{prog}(\hat{v}_i^r) \tag{10}$$

$$\max_{1 \le r \le t} \text{prog}(v_i^t) \le \max_{1 \le r < t} \max \left\{ \max_{1 \le i \le n} \text{prog}(\hat{v}_i^r), \text{prog}(\mathcal{Y}_i^r) \right\} \le \max_{1 \le r < t} \max_{1 \le i \le n} \text{prog}(\hat{v}_i^r) + 1$$

$$\text{prog}(\hat{x}^t) \le \max_{1 \le r \le t} \max_{1 \le i \le n} \text{prog}(\hat{v}_i^r)$$

Next, we outline the proofs for the lower bounds presented in Theorem 2. For each case, we provide separate proofs for terms in the lower bound by constructing different hard-to-optimize examples, respectively. The construction of these proofs follows four steps:

- Constructing a set of zero-chain local functions $\{f_i\}_{i=1}^n$.

- Constructing a set of independent unbiased compressors $\{C_i\}_{i=1}^n \subseteq \mathcal{U}_\omega^{\text{ind}}$. These compressors are delicately designed to impede algorithms from expanding the non-zero coordinates of model parameters.

- Establishing a limitation on zero-respecting algorithms that utilize the predefined compressor with $t$ rounds of compressed communication on each worker. This limitation is based on the non-zero coordinates of model parameters.

- Translating the above limitation into the lower bound of the complexity measure defined in equation (3).

While the overall proof structure is similar to that of [19], our novel construction of functions and compressors enable us to derive lower bounds for independent compressors. These lower bounds clarify the unique properties and benefits of independent compressors.

We will use the following lemma in the analysis of the third step.

**Lemma 3** ([19], Lemma 3). *Given a constant $p \in [0, 1]$ and random variables $\{B^t\}_{t=0}^\infty$ such that $B^t \le B^{(t-1)} + 1$ and $\mathbb{P}(B^t \le B^{t-1} \mid \{B^r\}_{r=0}^{t-1}) \ge 1 - p$ for any $t \ge 1$, it holds for $t \ge 1/p$, with probability at least $1 - e^{-1}$, that $B^t \le B^0 + ept$.*

## C.1 Strongly-convex case

Below, we present two examples, each of which corresponding to a lower bound $LB_m$ for $T_\epsilon$. We integrate the two lower bounds together and use the inequality

$$T_\epsilon \ge \max_{1 \le m \le 2} \{LB_m\} = \Omega\left(LB_1 + LB_2\right)$$

to accomplish the lower bound for strongly-convex problems in Theorem 2.

**Example 1.** In this example, we prove the lower bound $\Omega((1 + \omega)(1 + \sqrt{\kappa/n}) \ln(\mu\Delta/\epsilon))$.

(Step 1.) We assume the variable $x \in \ell_2 \triangleq \{([x]_1, [x]_2, \ldots,) : \sum_{r=1}^{\infty} [x]_r^2 < \infty\}$ to be infinitely dimensional and square-summable for simplicity. It is easy to adapt the argument for finitely dimensional variables as long as the dimension is proportionally larger than $t$. Let $M$ be

$$
M = \begin{bmatrix} 2 & -1 & & & \\ -1 & 2 & -1 & & \\ & -1 & 2 & -1 & \\ & & \ddots & \ddots & \ddots \end{bmatrix} \in \mathbb{R}^{\infty \times \infty},
$$

then it is easy to see $0 \preceq M \preceq 4I$. Let $\{f_i\}_{i=1}^n$ be as follows

$$
f_i(x) = \begin{cases} \frac{\mu}{2} \|x\|^2 + \frac{L-\mu}{4} \sum_{r \geq 0}([x]_{nr+i} - [x]_{nr+i+1})^2, & \text{if } 1 \leq i \leq n-1, \\ \frac{\mu}{2} \|x\|^2 + \frac{L-\mu}{4} \left([x]_1^2 + \sum_{r \geq 1}([x]_{nr} - [x]_{nr+1})^2 - 2\lambda[x]_1\right), & \text{if } i = n. \end{cases}
$$

where $\lambda \in \mathbb{R} \setminus \{0\}$ is to be specified. It is easy to see that $\sum_{r \geq 0}([x]_{nr+i} - [x]_{nr+i+1})^2$ and $[x]_1^2 + \sum_{r \geq 0}([x]_{nr} - [x]_{nr+1})^2 - 2\lambda[x]_1$ are convex and 4-smooth. Consequently, all $f_i$s are $L$-smooth and $\mu$-strongly convex. More importantly, it is easy to verify that all $f_i$s defined above are zero-chain functions and satisfy

$$
\text{prog}(\nabla f_i(x)) \begin{cases} = \text{prog}(x) + 1, & \text{if } \text{prog}(x) \equiv i \mod n, \\ \leq \text{prog}(x), & \text{otherwise.} \end{cases} \tag{11}
$$

We further have $f(x) = \frac{1}{n} \sum_{i=1}^n f_i(x) = \frac{\mu}{2} \|x\|^2 + \frac{L-\mu}{4n}(x^\top M x - 2\lambda[x]_1)$. For the functions defined above, we also establish that

**Lemma 4.** *Let $\kappa \triangleq L/\mu \geq 1$, it holds for any $x$ that,*

$$
f(x) - \min_x f(x) \geq \frac{\mu}{2} \left(1 - 2\left(1 + \sqrt{1 + \frac{2(\kappa-1)}{n}}\right)^{-1}\right)^{2\text{prog}(x)} \|x^0 - x^\star\|^2.
$$

*Proof.* The minimum $x^\star$ of function $f$ satisfies $\left(\frac{L-\mu}{2n} M + \mu\right) x^\star - \lambda \frac{L-\mu}{2} e_1 = 0$, which is equivalent to

$$
\frac{2\kappa + 2n - 2}{\kappa - 1}[x^\star]_1 - [x^\star]_2 = \lambda,
$$

$$
-[x^\star]_{j-1} + \frac{2\kappa + 2n - 2}{\kappa - 1}[x^\star]_j - [x^\star]_{j+1} = 0, \quad \forall j \geq 2. \tag{12}
$$

Note that

$$
q = \frac{\kappa + n - 1 - \sqrt{n(2\kappa + n - 2)}}{\kappa - 1} = 1 - \frac{2}{1 + \sqrt{1 + \frac{2(\kappa-1)}{n}}}
$$

is the only root of the equation $q^2 - \frac{2\kappa+2n-2}{\kappa-1}q + 1 = 0$ that is smaller than 1. Then it is straight forward to check $x^\star = \left([x^\star]_j = \lambda q^j\right)_{j \geq 1}$ satisfies (12). By the strong convexity of $f$, $x^\star$ is the unique solution. Therefore, we have that

$$
\|x - x^\star\|^2 \geq \sum_{j=\text{prog}(x)+1}^{\infty} \lambda^2 q^{2j} = \lambda^2 \frac{q^{2(r+1)}}{1-q^2} = q^{2r} \|x^0 - x^\star\|^2.
$$

Finally, using the strong convexity of $f$ leads to the conclusion. $\qquad \square$

Following the proof of Lemma 4, we have

$$
\|x^0 - x^\star\|^2 = \lambda^2 \sum_{j=1}^{\infty} q^{2j} = \lambda^2 \frac{q^2}{1-q^2}
$$

Therefore, for any given $\Delta > 0$, letting $\lambda = \sqrt{((1-q^2)\Delta)/q^2}$ results in $\|x^0 - x^\star\|^2 = \Delta$. Consequently, our construction ensures $\{f_i\}_{i=1}^n \in \mathcal{F}_{L,\mu}^\Delta$.

(Step 2.) For the construction of $\omega$-unbiased compressors, we consider $\{C_i\}_{i=1}^n$ to be independent random sparsification compressors. Building upon Example 1, we make a slight modification: during a round of communication on any worker, each coordinate is independetly chosen with a probability of $(1+\omega)^{-1}$ to be transmitted, and if selected, its value is scaled by $(1+\omega)$ and then the scaled value is transmitted. Notably, the indices of chosen coordinates are not identical across all workers due to the independence of compressors. It can be easily verified that this construction ensures that $\{C_i\}_{i=1}^n \subseteq \mathcal{U}_\omega^{\text{ind}}$.

(Step 3.) Since the algorithmic output $\hat{x}^t$ calculated by the server lies in the linear manifold spanned by received messages, we can use (10) to obtain the following expression:

$$\text{prog}(\hat{x}^t) \leq \max_{1 \leq r \leq t} \max_{1 \leq i \leq n} \max\{\text{prog}(u^r), \text{prog}(\hat{v}_i^r)\} = \max_{1 \leq r \leq t} \max_{1 \leq i \leq n} \text{prog}(\hat{v}_i^r) \triangleq B^t. \qquad (13)$$

We next bound $B^t$ with $B^0 := 0$ by showing that $\{B^t\}_{t=0}^\infty$ satisfies Lemma 3 with $p = (1+\omega)^{-1}$.

For any linear-spanning algorithm $A$, according to (11), the worker $i$ can only attain one additional non-zero coordinate through local gradient-based updates when $\text{prog}(\mathcal{Y}_i^t) \equiv i \mod n$. In other words, upon receiving messages $\{u_i^r\}_{r=1}^t$ from the server, we have

$$\text{prog}(v_i^t) \leq \begin{cases} \max_{1 \leq r \leq t} \text{prog}(u_i^r) + 1 \leq B^{t-1} + 1, & \text{if } \text{prog}(\mathcal{Y}_i^t) \equiv i \mod n, \\ \max_{1 \leq r \leq t} \text{prog}(u_i^r) \leq B^{t-1}, & \text{otherwise.} \end{cases}$$

Consequently, we have

$$\max_{1 \leq r \leq t} \text{prog}(v_i^r) \leq \max_{1 \leq r \leq t} B^{r-1} + 1 = B^{t-1} + 1.$$

It then follows from the definition of the constructed $C_i$ in Step 2 that $\max_{1 \leq i \leq n} \text{prog}(\hat{v}_i^t) \leq \max_{1 \leq i \leq n} \text{prog}(v_i^t)$, and therefore we have:

$$B^t \leq \max_{1 \leq r \leq t} \max_{1 \leq i \leq n} \text{prog}(v_i^r) \leq B^{t-1} + 1.$$

Next, we aim to prove that $B^t \leq B^{t-1} + 1$ with a probability of at least $\omega/(1+\omega)$. For any $t \geq 1$, let $i \in \{1, \ldots, n\}$ be such that $B^{t-1} \equiv i \mod n$. Due to the property in equation (11), during the $t$-th communication round, if $\text{prog}(\mathcal{Y}_i^t) = B^{t-1}$, worker $i$ can push the number of non-zero entries forward by 1, resulting in $\text{prog}(v_i^t) = B^{t-1} + 1$, using local gradient updates. Note that any other worker $j$ cannot achieve this even if $\text{prog}(\mathcal{Y}_j^t) = B^{t-1}$ due to equation (11).

Therefore, to achieve $B^t = B^{t-1} + 1$, it is necessary for worker $i$ to transmit a non-zero value at the $(B^{t-1} + 1)$-th entry to the server. Otherwise, we have $B^t \leq B^{t-1}$. However, since the compressor $C_i$ associated with worker $i$ has a probability $\omega/(1+\omega)$ to zero out the $(B^{t-1} + 1)$-th entry in the $t$-th communication round, we have

$$\mathbb{P}\left(B^t \leq B^{t-1} \mid \{B^r\}_{r=0}^{t-1}\right) \geq \omega/(1+\omega).$$

In summary, we have shown that $B^t \leq B^{t-1} + 1$ and $\mathbb{P}(B^t \leq B^{t-1} \mid \{B^r\}_{r=0}^{t-1}) \geq \omega/(1+\omega)$.

By applying Lemma 3, we can conclude that for any $t \geq (1+\omega)^{-1}$, with a probability of at least $1 - e^{-1}$, it holds that $B^t \leq et/(1+\omega)$ and hence $\text{prog}(\hat{x}^t) \leq et/(1+\omega)$ due to (13).

(Step 4.) Using Lemma 4 and that $\text{prog}(\hat{x}^t) \leq et/(1+\omega)$ with probability at least $1 - e^{-1}$, we obtain

$$\mathbb{E}[f(\hat{x}^t)] - \min_x f(x) \geq \frac{(1-e^{-1})\mu\Delta}{2}\left(1 - 2\left(1 + \sqrt{1 + \frac{2(\kappa-1)}{n}}\right)^{-1}\right)^{2et/(1+\omega)} \qquad (14)$$

$$= \Omega\left(\mu\Delta \exp\left(-\frac{4et}{(\sqrt{\kappa/n} + 1)(1+\omega)}\right)\right).$$

Therefore, to ensure $\mathbb{E}[f(\hat{x}^t)] - \min_x f(x) \leq \epsilon$, relation (14) implies the lower bound $T_\epsilon = \Omega((1+\omega)(1+\sqrt{\kappa/n})\ln(\mu\Delta/\epsilon))$.

**Example 2.** Considering $f_1 = f$ to be homogeneous and $C_i = I$ to be a loss-less compressor for all $1 \leq i \leq n$, the problem reduces to single-node convex optimization. In this case, the lower bound of $\Omega(\sqrt{\kappa}\ln(\mu\Delta/\epsilon))$ is well-known in the literature, as shown in [44, 43].

With the two lower bounds achieved in Examples 1 and 2, we have

$$T_\epsilon = \Omega\Big((1+\omega)(1+\sqrt{\kappa/n})\ln(\mu\Delta/\epsilon) + \sqrt{\kappa}\ln(\mu\Delta/\epsilon)\Big)$$

$$= \Omega\Big((1 + \omega + \sqrt{\kappa/n} + \omega\sqrt{\kappa/n} + \sqrt{\kappa})\ln(\mu\Delta/\epsilon)\Big)$$

$$= \Omega\Big((\omega + \omega\sqrt{\kappa/n} + \sqrt{\kappa})\ln(\mu\Delta/\epsilon)\Big)$$

which is the result for the strongly-convex case in Theorem 2.

## C.2 Generally-convex case

Below, we present three examples, each of which corresponding to a lower bound $LB_m$ for $T_\epsilon$. We integrate the three lower bounds together and use the inequality

$$T_\epsilon \geq \max_{1 \leq m \leq 3}\{LB_m\} = \Omega(LB_1 + LB_2 + LB_3)$$

to accomplish the lower bound for the generally-convex case in Theorem 2.

**Example 1.** In this example, we prove the lower bound $\Omega((1+\omega)(L\Delta/\epsilon)^{1/2})$.

(Step 1.) We assume variable $x \in \mathbb{R}^d$, where $d$ can be sufficiently large and will be determined later. Let $M$ denote

$$M = \begin{bmatrix} 2 & -1 & & & & \\ -1 & 2 & -1 & & & \\ & \ddots & \ddots & \ddots & & \\ & & -1 & 2 & -1 \\ & & & -1 & 2 \end{bmatrix} \in \mathbb{R}^{d \times d},$$

it is easy to verify $0 \preceq M \preceq 4I$. Similar to example 1 of the strongly-convex case, we consider

$$f_i(x) = \begin{cases} \frac{L}{4}\sum_{r \geq 0}([x]_{nr+i} - [x]_{nr+i+1})^2, & \text{if } 1 \leq i \leq n-1, \\ \frac{L}{4}\left([x]_1^2 + \sum_{r \geq 1}([x]_{nr} - [x]_{nr+1})^2 - 2\lambda[x]_1\right), & \text{if } i = n. \end{cases}$$

where $\lambda \in \mathbb{R}\backslash\{0\}$ is to be specified. It is easy to see that all $f_i$s are $L$-smooth. We further have $f(x) = \frac{1}{n}\sum_{i=1}^n f_i(x) = \frac{L}{4n}\left(x^\top M x - 2\lambda[x]_1\right)$. The $f_i$ functions defined above are also zero-chain functions satisfying (11).

Following [43], it is easy to verify that the optimum of $f$ satisfies

$$x^\star = \left(\lambda\left(1 - \frac{k}{d+1}\right)\right)_{1 \leq k \leq d} \quad \text{and} \quad f(x^\star) = \min_x f(x) = -\frac{\lambda^2 L d}{4n(d+1)}.$$

More generally, it holds for any $0 \leq k \leq d$ that

$$\min_{x:\, \text{prog}(x) \leq k} f(x) = -\frac{\lambda^2 L k}{4n(k+1)}. \tag{15}$$

Since $\|x^0 - x^\star\|^2 = \frac{\lambda^2}{(d+1)^2}\sum_{k=1}^d k^2 = \frac{\lambda^2 d(2d+1)}{6(d+1)} \leq \frac{\lambda^2 d}{3}$, letting $\lambda = \sqrt{3\Delta/d}$, we have $\{f_i\}_{i=1}^n \in \mathcal{F}_{L,0}^\Delta$.

(Step 2.) Same as Step 2 of Example 1 of the strongly-convex case, we consider $\{C_i\}_{i=1}^n$ to be independent random sparsification operators.

(Step 3.) Following the same argument as step 3 of example 1 of the strongly-convex case, we have that for any $t \geq (1 + \omega)^{-1}$, it holds with probability at least $1 - e^{-1}$ that $\text{prog}(\hat{x}^t) \leq et/(1 + \omega)$.

(Step 4.) Thus, combining (15), we have

$$\mathbb{E}[f(\hat{x}^t)] - \min_x f(x) \geq (1 - e^{-1})\frac{\lambda^2 L}{4n}\left(\frac{d}{d+1} - \frac{et/(1+\omega)}{1 + et/(1+\omega)}\right)$$

$$= (1 - e^{-1})\frac{3L\Delta}{4nd}\left(\frac{d}{d+1} - \frac{et/(1+\omega)}{1 + et/(1+\omega)}\right)$$

Letting $d = 1 + et/(1 + \omega)$, we further have

$$\mathbb{E}[f(\hat{x}^t)] - \min_x f(x) \geq \frac{3(1 - e^{-1})L\Delta}{8net(1+\omega)^{-1}(1 + 2et(1+\omega)^{-1})} = \Omega\left(\frac{(1+\omega)^2 L\Delta}{nt^2}\right).$$

Therefore, to ensure $\mathbb{E}[f(\hat{x}^t)] - \min_x f(x) \leq \epsilon$, the above inequality implies the lower bound to be $T = \Omega((1 + \omega)(L\Delta/(n\epsilon))^{\frac{1}{2}})$.

**Example 2.** Considering $f_1 = f$ to be homogeneous and $C_i = I$ to be a loss-less compressor for all $1 \leq i \leq n$. The problem reduces to the single-node convex optimization. The lower bound $\Omega(\sqrt{L\Delta/\epsilon})$ is well-known in literature, see, *e.g.*, [44, 43].

**Example 3.** In this example, we prove the lower bound $\Omega(\omega \ln(L\Delta/\epsilon))$.

(Step 1.) We consider $f_1 = \cdots = f_{n-1} = L\|x\|^2/2$ and $f_n = L\|x\|^2/2 + n\lambda\langle \mathbb{1}_d, x\rangle$ where $\mathbb{1}_d \in \mathbb{R}^d$ is the vector with all enries being 1 and $\lambda \in \mathbb{R}$ is to be determined. By definition, $\{f_i\}_{i=1}^n$ are $\mu$-strongly-convex and $L$-smooth and the solution $x^\star = -\frac{\lambda}{L}\mathbb{1}_d$. Letting $\lambda = L\sqrt{\Delta}/\sqrt{n}$, we have $\|x^\star - x^0\|^2 = \Delta$. Thus, the construction ensures $\{f_i\}_{i=1}^n \in \mathcal{F}_{L,\mu}^\Delta$.

(Step 2.) Same as in Example 1, we consider $\{C_i\}_{i=1}^n$ to be independent random sparsification operators.

(Step 3.) By the construction of $\{f_i\}_{i=1}^n$, we observe that the optimization process relies solely on transmitting the information of $\mathbb{1}_d$ from worker $n$ to the server. Let $E^t$ denote the set of entries at which the server has received a non-zero value from worker $n$ in the first $t$ communication rounds. Note that for each entry, due to the construction of $\{C_i\}_{i=1}^n$, the server has a probability of at least $(\omega/(1+\omega))^t$ of not receiving a non-zero value at that entry from worker $n$. Consequently, $|(E^t)^c|$ is lower bounded by the sum of $n$ independent $\text{Bernoulli}(\omega^t/(1+\omega)^t)$ random variables. Therefore, we have $\mathbb{E}[|(E^t)^c|] \geq d\omega^t/(1+\omega)^t$.

(Step 4.) Given $|E^t|$, due to the linear-spanning property, we have $\hat{x}^t \in \text{span}\{e_j : j \in E^t\}$ where $e_j$ is the $j$-th canonical vector. As a result, we have

$$\mathbb{E}[f(\hat{x}^t)] - \min_x f(x)$$

$$\geq \mathbb{E}[\min_{x \in \text{span}\{e_j : j \in E^t\}} f(x)] - \min_x f(x) = \frac{L\Delta}{2}\frac{\mathbb{E}[|(E^t)^c|]}{d} \geq \frac{L\Delta}{2}\frac{\omega^t}{(1+\omega)^t}. \quad (16)$$

Therefore, to ensure $\mathbb{E}[f(\hat{x}^t)] - \min_x f(x) \leq \epsilon$, (16) implies the lower bound $T_\epsilon = \Omega(\omega \ln(L\Delta/\epsilon))$.

With the three lower bounds achieved in Examples 1, 2, and 3, we have

$$T_\epsilon = \Omega\left(\sqrt{\frac{L\Delta}{\epsilon}} + (1 + \omega)\sqrt{\frac{L\Delta}{n\epsilon}} + \omega \ln(L\Delta/\epsilon)\right)$$

$$= \Omega\left(\sqrt{\frac{L\Delta}{\epsilon}} + \omega\sqrt{\frac{L\Delta}{n\epsilon}} + \omega \ln(L\Delta/\epsilon)\right)$$

which is the result for the generally-convex case in Theorem 2.

# D    Proof of Theorem 3

## D.1    Strongly-convex case

We first present several important lemmas, followed by the definition of a Lyapunov function with delicately chosen coefficients for each term. Finally, we prove Theorem 3 by utilizing these lemmas. Throughout the convergence analysis, we use the following notations:

$$\mathcal{W}^k = f(w^k) - f^\star, \quad \mathcal{Y}^k = f(y^k) - f^\star, \quad \mathcal{Z}^k = \|z^k - x^\star\|^2,$$

$$\mathcal{H}^k = \frac{1}{n}\sum_{i=1}^n \|h_i^k - \nabla f_i(w^k)\|^2, \quad \mathcal{G}^k = \|g^k - \nabla f(x^k)\|^2,$$

$$\mathcal{G}_w^k = \frac{1}{n}\sum_{i=1}^n \|\nabla f_i(w^k) - \nabla f_i(x^k)\|^2, \quad \mathcal{G}_y^k = \frac{1}{n}\sum_{i=1}^n \|\nabla f_i(y^k) - \nabla f_i(x^k)\|^2.$$

We use $\mathbb{E}_k$ or $\mathbb{E}$ indicate the expectation with respect to the randomness in the $k$-th iteration or all historical randomness, respectively.

**Lemma 5.** *If $0 \le \beta \le 1$, it holds for $\forall\, k \ge 0$ that,*

$$\mathcal{Z}^{k+1} \le 2\gamma_k\langle g^k, x^\star - x^k\rangle + \frac{2\gamma_k\beta\theta_2}{\theta_{1,k}}\langle g^k, w^k - x^k\rangle + \frac{2\gamma_k\beta(1 - \theta_{1,k} - \theta_2)}{\theta_{1,k}}\langle g^k, y^k - x^k\rangle$$

$$+ \beta\mathcal{Z}^k + (1 - \beta)\|x^k - x^\star\|^2 + \gamma_k^2\|g^k\|^2. \tag{17}$$

*Proof.* Following the update rules in Algorithm 1, we have

$$\mathcal{Z}^{k+1} = \left\|\beta z^k + (1 - \beta)x^k - x^\star + \frac{\gamma_k}{\eta_k}(y^{k+1} - x^k)\right\|^2$$

$$= \|\beta(z^k - x^\star) + (1 - \beta)(x^k - x^\star)\|^2 + \gamma_k^2\|g^k\|^2$$

$$+ \langle 2\gamma_k g^k, \beta z^k + (1 - \beta)x^k - x^\star\rangle. \tag{18}$$

Since $x^k = \theta_{1,k}z^k + \theta_2 w^k + (1 - \theta_{1,k} - \theta_2)y^k$, we have

$$\beta z^k + (1 - \beta)x^k - x^\star = (x^k - x^\star) + \frac{\beta\theta_2}{\theta_{1,k}}(x^k - w^k) + \frac{\beta(1 - \theta_{1,k} - \theta_2)}{\theta_{1,k}}(x^k - y^k). \tag{19}$$

Plugging (19) into (18), using

$$\|\beta(z^k - x^\star) + (1 - \beta)(x^k - x^\star)\|^2 \le \beta\|z^k - x^\star\|^2 + (1 - \beta)\|x^k - x^\star\|^2,$$

we obtain (17).    $\square$

**Lemma 6.** *Under Assumption 1, if parameters satisfy $\theta_{1,k}, \theta_2, 1 - \theta_{1,k} - \theta_2 \in (0, 1)$, $\eta_k \in \left(0, \frac{1}{2L}\right]$, $\gamma_k = \frac{\eta_k}{2\theta_{1,k} + \eta_k\mu}$ and $\beta = 1 - \gamma\mu = \frac{2\theta_{1,k}}{2\theta_{1,k} + \eta_k\mu}$, then we have for any iteration $k \ge 0$ that*

$$\frac{2\gamma_k\beta}{\theta_{1,k}}\mathbb{E}_k[\mathcal{Y}^{k+1}] + \mathbb{E}_k[\mathcal{Z}^{k+1}] \le \frac{2\gamma_k\beta\theta_2}{\theta_{1,k}}\mathcal{W}^k + \frac{2\gamma_k\beta(1 - \theta_{1,k} - \theta_2)}{\theta_{1,k}}\mathcal{Y}^k + \beta\mathcal{Z}^k + \frac{5\gamma_k\beta\eta_k}{4\theta_{1,k}}\mathcal{G}^k$$

$$- \frac{\gamma_k\beta\theta_2}{L\theta_{1,k}}\mathcal{G}_w^k - \frac{\gamma_k\beta(1 - \theta_{1,k} - \theta_2)}{L\theta_{1,k}}\mathcal{G}_y^k. \tag{20}$$

*Proof.* By Assumption 1 and update rules in Algorithm 1, we have

$$f(y^{k+1}) \le f(x^k) + \langle \nabla f(x^k), y^{k+1} - x^k\rangle + \frac{L}{2}\|y^{k+1} - x^k\|^2$$

$$= f(x^k) - \langle \nabla f(x^k), \eta_k g^k\rangle + \frac{L}{2}\eta_k^2\|g^k\|^2$$

$$= f(x^k) - \eta_k\langle \nabla f(x^k) - g^k, g^k\rangle + \left(\frac{L\eta_k^2}{2} - \eta_k\right)\|g^k\|^2. \tag{21}$$

By $L$-smoothness and $\mu$-strongly convexity, we have for $\forall u \in \mathbb{R}^d$ that

$$f(u) \geq f(x^k) + \langle \nabla f(x^k), u - x^k \rangle + \frac{\mu}{2}\|u - x^k\|^2,$$

and that

$$f_i(u) \geq f_i(x^k) + \langle \nabla f_i(x^k), u - x^k \rangle + \frac{1}{2L}\|\nabla f_i(u) - \nabla f_i(x^k)\|^2,$$

thus we obtain for $\forall u \in \mathbb{R}^d$,

$$\begin{aligned}
f(x^k) \leq & f(u) - \langle \nabla f(x^k), u - x^k \rangle \\
& - \max\left\{ \frac{\mu}{2}\|u - x^k\|^2, \frac{1}{2Ln}\sum_{i=1}^{n}\|\nabla f_i(u) - \nabla f_i(x^k)\|^2 \right\}.
\end{aligned} \tag{22}$$

Applying Young's inequality to (21) and using $\eta_k \leq 1/(2L)$, we reach

$$\begin{aligned}
f(y^{k+1}) \leq & f(x^k) + \frac{\eta_k}{2}\mathcal{G}^k - \frac{\eta_k}{2}(1 - L\eta_k)\|g^k\|^2 \\
\leq & f(x^k) + \frac{\eta_k}{2}\mathcal{G}^k - \frac{\eta_k}{4}\|g^k\|^2.
\end{aligned} \tag{23}$$

Adding (17) in Lemma 5 to $\left(\frac{2\gamma_k\beta}{\theta_{1,k}} + 2\gamma_k(1-\beta)\right) \times$(23) + $2\gamma_k \times$(22) (where $u = x^\star$) + $\frac{2\gamma_k\beta\theta_2}{\theta_{1,k}} \times$(22) (where $u = w^k$) + $\frac{2\gamma_k\beta(1-\theta_{1,k}-\theta_2)}{\theta_{1,k}} \times$(22) (where $u = y^k$) and using the unbiasedness of $g^k$, we obtain

$$\frac{2\gamma_k\beta}{\theta_{1,k}}\mathbb{E}_k[\mathcal{Y}^{k+1}] + \mathbb{E}_k[\mathcal{Z}^{k+1}]$$

$$\begin{aligned}
\leq & \beta\mathcal{Z}^k + (1 - \beta - \mu\gamma_k)\|x^k - x^\star\|^2 + \left(\gamma_k^2 - \frac{\eta_k\gamma_k\beta}{2\theta_{1,k}}\right)\mathbb{E}_k[\|g^k\|^2] + \eta_k\left(\frac{\gamma_k\beta}{\theta_{1,k}} + \gamma_k(1-\beta)\right)\mathcal{G}^k \\
& - \frac{\gamma_k\beta\theta_2}{L\theta_{1,k}}\mathcal{G}_w^k - \frac{\gamma_k\beta(1-\theta_{1,k}-\theta_2)}{L\theta_{1,k}}\mathcal{G}_y^k + \frac{2\gamma_k\beta\theta_2}{\theta_{1,k}}\mathcal{W}^k + \frac{2\gamma_k\beta(1-\theta_{1,k}-\theta_2)}{\theta_{1,k}}\mathcal{Y}^k \\
& - 2\gamma_k(1-\beta)\mathbb{E}_k[\mathcal{Y}^{k+1}] - \frac{\eta_k\gamma_k(1-\beta)}{2}\mathbb{E}_k[\|g^k\|^2]
\end{aligned}$$

On top of that, by applying our choice of the parameters, it can be easily verified that $1 - \beta - \mu\gamma_k = 0$, $\gamma_k^2 - \frac{\eta_k\gamma_k\beta}{2\theta_{1,k}} = 0$, $1 - \beta \leq \frac{\beta}{4\theta_{1,k}}$, which leads to (20). $\qquad\square$

**Lemma 7** ([32], Lemma 3, 4, 5). *Under Assumptions 1, 2, and 3, the iterates of Algorithm 1 satisfy the following inequalities:*

$$\mathbb{E}[\mathcal{W}^{k+1}] = (1-p)\mathbb{E}[\mathcal{W}^k] + p\mathbb{E}[\mathcal{Y}^k], \tag{24}$$

$$\mathbb{E}[\mathcal{G}^k] \leq \frac{2\omega}{n}\mathbb{E}[\mathcal{G}_w^k] + \frac{2\omega}{n}\mathbb{E}[\mathcal{H}^k], \tag{25}$$

$$\mathbb{E}[\mathcal{H}^{k+1}] \leq \left(1 - \frac{\alpha}{2}\right)\mathbb{E}[\mathcal{H}^k] + 2p\left(1 + \frac{2p}{\alpha}\right)(\mathbb{E}[\mathcal{G}_w^k] + \mathbb{E}[\mathcal{G}_y^k]). \tag{26}$$

Now we define a Lyapunov function $\Psi^k$ for $k \geq 1$ as

$$\Psi^k = \lambda_{k-1}\mathcal{W}^k + \frac{2\gamma_{k-1}\beta}{\theta_{1,k-1}}\mathcal{Y}^k + \mathcal{Z}^k + \frac{10\eta_{k-1}\omega(1+\omega)\gamma_{k-1}\beta}{\theta_{1,k-1}n}\mathcal{H}^k, \quad \forall k \geq 1, \tag{27}$$

where $\lambda_k = \frac{\gamma_k\beta}{p\theta_{1,k}}(\theta_{1,k} + \theta_2 - p + \sqrt{(p - \theta_{1,k} - \theta_2)^2 + 4p\theta_2})$. Furthermore, it is straightforward to verify that

$$\frac{2\gamma_k\beta\theta_2}{p\theta_{1,k}} \leq \lambda_k \leq \frac{2\gamma_k\beta(\theta_{1,k} + \theta_2)}{p\theta_{1,k}}.$$

Now we restate the convergence result in the strongly-convex case in Theorem 3 and prove it using Lemma 6, 7 and the Lyapunov function.

**Theorem 4.** *If $\mu > 0$ and parameters satisfy $\eta_k \equiv \eta = n\theta_2/(120\omega L)$, $\theta_{1,k} \equiv \theta_1 = 1/(3\sqrt{\kappa})$, $\alpha = p = 1/(1+\omega)$, $\gamma_k \equiv \gamma = \eta/(2\theta_1 + \eta\mu)$, $\beta = 2\theta_1/(2\theta_1 + \eta\mu)$, and $\theta_2 = 1/(3\sqrt{n} + 3n/\omega)$, then the number of communication rounds performed by ADIANA to find an $\epsilon$-accurate solution such that $\mathbb{E}[f(\hat{x})] - \min_x f(x) \leq \epsilon$ is at most $\mathcal{O}((\omega + (1 + \omega/\sqrt{n})\sqrt{\kappa})\ln(L\Delta/\epsilon))$.*

*Proof.* In the strongly-convex case, parameters $\{\gamma_k\}_{k \geq 1}$ and $\{\theta_{1,k}\}_{k \geq 1}$ are constants, then so is $\lambda_k$. Thus, we simply write $\gamma \triangleq \gamma_k$, $\theta_1 \triangleq \theta_{1,k}$, and $\lambda \triangleq \lambda_k$ for all $k \geq 1$. Considering $(20)+\lambda(24)+\frac{5\gamma\beta\eta}{4\theta_1}(25)+\frac{10\eta\omega(1+\omega)\gamma\beta}{n\theta_1}(26)$, we have

$$
\begin{aligned}
\mathbb{E}[\Psi^{k+1}] \leq{} & \left(\frac{2\gamma\beta\theta_2}{\theta_1} + (1-p)\lambda\right)\mathcal{W}^k + \left(\frac{2\gamma\beta(1-\theta_1-\theta_2)}{\theta_1} + p\lambda\right)\mathcal{Y}^k + \beta\mathcal{Z}^k \\
& + \left(1 - \frac{1}{4(1+\omega)}\right)\frac{10\eta\omega(1+\omega)\gamma\beta}{\theta_1 n}\mathcal{H}^k - \left(\frac{\gamma\beta\theta_2}{L\theta_1} - \frac{125\gamma\beta\eta\omega}{2n\theta_1}\right)\mathcal{G}_w^k \\
& - \left(\frac{\gamma\beta(1-\theta_1-\theta_2)}{L\theta_1} - \frac{60\eta\omega\gamma\beta}{n\theta_1}\right)\mathcal{G}_y^k.
\end{aligned}
\tag{28}
$$

By the definition of $\lambda$, we have

$$
\begin{aligned}
\frac{2\gamma\beta\theta_2}{\theta_1} + (1-p)\lambda ={} & \lambda\left(1 - p + \frac{2p\theta_2}{\sqrt{(p-\theta_1-\theta_2)^2 + 4p\theta_2} + \theta_1 + \theta_2 - p}\right) \\
={} & \lambda\left(1 - p + \frac{2p\theta_2}{2\theta_2 + \frac{4\theta_1\theta_2}{\sqrt{(p-\theta_1-\theta_2)^2+4p\theta_2}-\theta_1+\theta_2+p}}\right) \\
\leq{} & \lambda\left(1 - p + \frac{p}{1 + \frac{2\theta_1}{(p+\theta_1+\theta_2)-\theta_1+\theta_2+p}}\right) = \left(1 - \frac{p\theta_1}{p+\theta_1+\theta_2}\right)\lambda, \quad (29)
\end{aligned}
$$

and

$$
\begin{aligned}
\frac{2\gamma\beta(1-\theta_1-\theta_2)}{\theta_1} + p\lambda ={} & \frac{2\gamma\beta}{\theta_1}\left[1 - \theta_1 - \theta_2 + \frac{1}{2}\left(\theta_1 + \theta_2 - p + \sqrt{(p-\theta_1-\theta_2)^2+4p\theta_2}\right)\right] \\
={} & \frac{2\gamma\beta}{\theta_1}\left(1 - \frac{2p\theta_1}{p+\theta_1+\theta_2+\sqrt{(p-\theta_1-\theta_2)^2+4p\theta_2}}\right) \\
\leq{} & \left(1 - \frac{p\theta_1}{p+\theta_1+\theta_2}\right)\frac{2\gamma\beta}{\theta_1}.
\end{aligned}
\tag{30}
$$

From the choice of $\eta$, it is easy to verify that

$$
\frac{\gamma\beta\theta_2}{L\theta_1} - \frac{5\gamma\beta\eta\omega}{2n\theta_1} - \frac{60\eta\omega\gamma\beta}{n\theta_1} \geq 0,
\tag{31}
$$

and further noting $1 - \theta_1 - \theta_2 \geq \theta_2$,

$$
\frac{\gamma\beta(1-\theta_1-\theta_2)}{L\theta_1} - \frac{60\eta\omega\gamma\beta}{n\theta_1} \geq 0.
\tag{32}
$$

Plugging (29), (30), (31), and (32) into (28), we obtain

$$
\begin{aligned}
\mathbb{E}[\Psi^{k+1}] \leq{} & \left(1 - \min\left\{\frac{p\theta_1}{p+\theta_1+\theta_2}, \frac{\eta\mu}{2\theta_1+\eta\mu}, \frac{1}{4(1+\omega)}\right\}\right)\Psi^k \\
\leq{} & \left(1 - \frac{1}{\frac{p+\theta_1+\theta_2}{p\theta_1} + \frac{2\theta_1+\eta\mu}{\eta\mu} + 4(1+\omega)}\right)\Psi^k \\
\leq{} & \left(1 - \frac{1}{250\left(\omega + \left(1+\frac{\omega}{\sqrt{n}}\right)\sqrt{\kappa}\right)}\right)\Psi^k, \quad \forall k \geq 0,
\end{aligned}
\tag{33}
$$

where $\Psi^0 := \lambda \mathcal{W}^0 + \frac{2\gamma\beta}{\theta_1}\mathcal{Y}^0 + \mathcal{Z}^0 + \frac{10\eta\omega(1+\omega)\gamma\beta}{\theta_1 n}\mathcal{H}^0$. Note that since we use initialization $y^0 = z^0 = w^0 = h_i^0 = h^0, \forall 1 \le i \le n$, we have $\mathcal{W}^0 = \mathcal{Y}^0 \le (L\Delta)/2$, $\mathcal{Z}^0 \le \Delta$, $\mathcal{H}^0 \le L^2\Delta$, which indicates that

$$\Psi^0 \le \frac{L}{2} \cdot (\lambda_W + \lambda_Y + \lambda_Z + \lambda_H)\Delta,$$

where $\lambda_W = \lambda \ge \frac{2\gamma\beta\theta_2}{\theta_1 p}$, $\lambda_Y = \frac{2\gamma\beta}{\theta_1}$, $\lambda_Z = \frac{2}{L}$, $\lambda_H = \frac{20\eta\omega(1+\omega)\gamma\beta L}{\theta_1 n}$. These coefficients have the following inequalities:

$$\lambda_W + \lambda_Y \ge \frac{4\eta(\theta_2 + p)}{p(2\theta_1 + \eta\mu)^2} = \frac{n\theta_2(\theta_2 + p)}{30\omega Lp(2/3\sqrt{\kappa} + n\theta_2/120\omega\kappa)^2} \ge \frac{n\theta_2(\theta_2 + p)\kappa}{15\omega Lp}$$

$$\ge \frac{\kappa}{135L} \ge \frac{1}{270}\lambda_Z,$$

and

$$\frac{3}{32}(\lambda_W + \lambda_Y) \ge \frac{\kappa}{1440L} \ge \frac{(1+\omega)n\theta_2^2\kappa}{160\omega L} \ge \frac{40\eta^2\omega(1+\omega)L}{(2\theta_1 + \eta\mu)^2 n} = \lambda_H.$$

Consequently, the initial value of the Lyapunov function can be bounded as

$$\Psi^0 \le 136L(\lambda_W + \lambda_Y)\Delta,$$

which together with (33) further implies that

$$\min\{\mathbb{E}[f(w^T)], \mathbb{E}[f(y^T)]\} - f^\star$$

$$\le \min\left\{\frac{1}{\lambda_W}, \frac{1}{\lambda_Y}\right\}\left(1 - \frac{1}{250\left(\omega + \left(1 + \frac{\omega}{\sqrt{n}}\right)\sqrt{\kappa}\right)}\right)^T \Psi^0$$

$$\le 272L\Delta\left(1 - \frac{1}{250\left(\omega + \left(1 + \frac{\omega}{\sqrt{n}}\right)\sqrt{\kappa}\right)}\right)^T.$$

Thus, $\mathcal{O}\left(\left(\omega + \left(1 + \frac{\omega}{\sqrt{n}}\right)\sqrt{\kappa}\right)\ln\left(\frac{L\Delta}{\epsilon}\right)\right)$ iterations are sufficient to guarantee an $\epsilon$-solution. $\qquad\square$

### D.2 Generally-convex case

In this subsection, we restate the convergence result in the generally-convex case as in Theorem 3 and prove it using Lemma 6, 7 and the Lyapunov function defined in (27).

**Theorem 5.** *If $\mu = 0$ and parameters satisfy $\alpha = 1/(1 + \omega)$, $\beta = 1$, $p = \theta_2 = 1/(3(1 + \omega))$, $\theta_{1,k} = 9/(k + 27(1 + \omega))$, $\gamma_k = \eta_k/(2\theta_{1,k})$, and*

$$\eta_k = \min\left\{\frac{k + 1 + 27(1 + \omega)}{9(1 + \omega)^2(1 + 27(1 + \omega))L}, \frac{3n}{200\omega(1 + \omega)L}, \frac{1}{2L}\right\},$$

*then the number of communication rounds performed by ADIANA to find an $\epsilon$-accurate solution such that $\mathbb{E}[f(\hat{x})] - \min_x f(x) \le \epsilon$ is provided by $\mathcal{O}((1 + \omega/\sqrt{n})\sqrt{L\Delta/\epsilon} + (1 + \omega)\sqrt[3]{L\Delta/\epsilon})$.*

*Proof.* Considering $(20) + \lambda_k(24) + \frac{5\gamma_k\beta\eta_k}{4\theta_{1,k}}(25) + \frac{10\eta_k\omega(1+\omega)\gamma_k\beta}{n\theta_{1,k}}(26)$ and applying the choice of $\theta_2$, $p$ and $\alpha$, we have

$$\mathbb{E}_k[\Psi^{k+1}]$$

$$\le \left(\frac{2\gamma_k\beta\theta_2}{\theta_{1,k}} + (1 - p)\lambda_k\right)\mathcal{W}^k + \left(\frac{2\gamma_k\beta(1 - \theta_{1,k} - \theta_2)}{\theta_{1,k}} + p\lambda_k\right)\mathcal{Y}^k + \beta\mathcal{Z}^k$$

$$+ \left(1 - \frac{1}{4(1+\omega)}\right)\frac{10\eta_k\omega(1+\omega)\gamma_k\beta}{n\theta_{1,k}}\mathcal{H}^k - \left(\frac{\gamma_k\beta\theta_2}{L\theta_{1,k}} - \frac{5\omega\gamma_k\beta\eta_k}{2n\theta_{1,k}} - \frac{100\eta_k\omega\gamma_k\beta}{9n\theta_{1,k}}\right)\mathcal{G}_w^k$$

$$- \left(\frac{\gamma_k\beta(1 - \theta_{1,k} - \theta_2)}{L\theta_{1,k}} - \frac{100\eta_k\omega\gamma_k\beta}{9n\theta_{1,k}}\right)\mathcal{G}_y^k. \tag{34}$$

Similar to the proof of Theorem 4, we can simplify (34) by validating

$$
\begin{cases}
\frac{2\gamma_k\beta\theta_2}{\theta_{1,k}} + (1-p)\lambda_k \leq \left(1 - \frac{p\theta_{1,k}}{p+\theta_{1,k}+\theta_2}\right)\lambda_k \leq \left(1 - \frac{\theta_{1,k}}{3}\right)\lambda_k, \\
\frac{2\gamma_k\beta(1-\theta_{1,k}-\theta_2)}{\theta_{1,k}} + p\lambda_k \leq \left(1 - \frac{p\theta_{1,k}}{p+\theta_{1,k}+\theta_2}\right)\frac{2\gamma_k\beta}{\theta_{1,k}} \leq \left(1 - \frac{\theta_{1,k}}{3}\right)\frac{2\gamma_k\beta}{\theta_{1,k}}, \\
\frac{\gamma_k\beta\theta_2}{L\theta_{1,k}} - \frac{5\omega\gamma_k\beta\eta_k}{2n\theta_{1,k}} - \frac{100\eta_k\omega\gamma_k\beta}{9n\theta_{1,k}} \geq 0, \\
\frac{\gamma_k\beta(1-\theta_{1,k}-\theta_2)}{L\theta_{1,k}} - \frac{100\eta_k\omega\gamma_k\beta}{9n\theta_{1,k}} \geq 0,
\end{cases}
$$

and then obtain

$$
\mathbb{E}_k[\Psi^{k+1}] \leq \left(1 - \frac{\theta_{1,k}}{3}\right)\lambda_k\mathcal{W}^k + \left(1 - \frac{\theta_{1,k}}{3}\right)\frac{2\gamma_k}{\theta_{1,k}}\mathcal{Y}^k + \mathcal{Z}^k
$$
$$
+ \left(1 - \frac{1}{4(1+\omega)}\right)\frac{10\eta_k\omega(1+\omega)\gamma_k}{\theta_{1,k}n}\mathcal{H}^k. \tag{35}
$$

For $\forall k \geq 1$, we have $\theta_{1,k} \leq \theta_{1,k-1}$ and thus

$$
\left(1 - \frac{\theta_{1,k}}{3}\right)\lambda_k = \left(1 - \frac{3}{k+27(1+\omega)}\right)\frac{\eta_k}{2p\theta_{1,k}^2}\left(\theta_{1,k} + \sqrt{\theta_{1,k}^2 + 4p\theta_2}\right)
$$
$$
\leq \left(1 - \frac{3}{k+27(1+\omega)}\right)\frac{\eta_k}{2p\theta_{1,k}^2}(\theta_{1,k-1} + \sqrt{\theta_{1,k-1}^2 + 4p\theta_2})
$$
$$
= \left(1 - \frac{3}{k+27(1+\omega)}\right)\left(\frac{k+27(1+\omega)}{k-1+27(1+\omega)}\right)^2\frac{\eta_k}{\eta_{k-1}}\lambda_{k-1}.
$$

Further noting $\frac{\eta_k}{\eta_{k-1}} \leq 1 + \frac{1}{k+27(1+\omega)}$, we obtain

$$
\left(1 - \frac{\theta_{1,k}}{3}\right)\lambda_k \leq \left(1 - \frac{3}{k+27(1+\omega)}\right)\left(1 - \frac{1}{k+27(1+\omega)}\right)^{-2}\left(1 + \frac{1}{k+27(1+\omega)}\right)\lambda_{k-1}
$$

$$
\leq \lambda_{k-1}. \tag{36}
$$

Similarly,

$$
\left(1 - \frac{\theta_{1,k}}{3}\right)\frac{2\gamma_k}{\theta_{1,k}}
$$
$$
= \left(1 - \frac{3}{k+27(1+\omega)}\right)\left(\frac{k+27(1+\omega)}{k-1+27(1+\omega)}\right)^2\frac{\eta_k}{\eta_{k-1}}\frac{2\gamma_{k-1}}{\theta_{1,k-1}}
$$
$$
\leq \left(1 - \frac{3}{k+27(1+\omega)}\right)\left(1 - \frac{1}{k+27(1+\omega)}\right)^{-2}\left(1 + \frac{1}{k+27(1+\omega)}\right)\frac{2\gamma_{k-1}}{\theta_{1,k-1}}
$$
$$
\leq \frac{2\gamma_{k-1}}{\theta_{1,k-1}}, \tag{37}
$$

and

$$
\left(1 - \frac{1}{4(1+\omega)}\right)\frac{10\eta_k\omega(1+\omega)\gamma_k\beta}{n\theta_{1,k}}
$$
$$
= \left(1 - \frac{1}{4(1+\omega)}\right)\left(\frac{k+27(1+\omega)}{k-1+27(1+\omega)}\right)^2\left(\frac{\eta_k}{\eta_{k-1}}\right)^2\frac{10\eta_{k-1}\omega(1+\omega)\gamma_{k-1}\beta}{n\theta_{1,k-1}}
$$
$$
\leq \frac{\left(1 - \frac{5}{k+27(1+\omega)}\right)\left(1 + \frac{3}{k+27(1+\omega)}\right)}{\left(1 - \frac{1}{k+27(1+\omega)}\right)^2}\frac{10\eta_{k-1}\omega(1+\omega)\gamma_{k-1}\beta}{n\theta_{1,k-1}}
$$
$$
\leq \frac{10\eta_{k-1}\omega(1+\omega)\gamma_{k-1}\beta}{\theta_{1,k-1}n}. \tag{38}
$$

Combining (35),(36),(37), and (38), we have for $\forall\, k \geq 1$ that

$$\mathbb{E}_k[\Psi^{k+1}] \leq \Psi^k. \tag{39}$$

By applying (39) with $k = T-1, T-2, \cdots, 1$ and (35) with $k = 0$, we obtain

$$\mathbb{E}[\Psi^T] \leq \left(1 - \frac{\theta_{1,0}}{3}\right)\frac{2\gamma_0(\theta_{1,0}+\theta_2)}{p\theta_{1,0}}\mathcal{W}^0 + \left(1 - \frac{\theta_{1,0}}{3}\right)\frac{2\gamma_0}{\theta_{1,0}}\mathcal{Y}^0 + \mathcal{Z}^0$$

$$+ \left(1 - \frac{1}{4(1+\omega)}\right)\frac{10\eta_0\omega(1+\omega)\gamma_0}{\theta_{1,0}n}\mathcal{H}^0$$

$$\leq \frac{2}{L}\mathcal{W}^0 + \frac{1}{L}\mathcal{Y}^0 + \mathcal{Z}^0 + \frac{3}{40L^2}\mathcal{H}^0 \leq \left(1 + \frac{1}{2} + 1 + \frac{3}{40}\right)\Delta \leq 3\Delta.$$

Note that

$$\Psi^T \geq \lambda_{T-1}\mathcal{W}^T + \frac{2\gamma_{T-1}\beta}{\theta_{1,T-1}}\mathcal{Y}^T \geq \frac{2\gamma_{T-1}\beta\theta_2}{\theta_{1,T-1}p}\mathcal{W}^T + \frac{2\gamma_{T-1}\beta}{\theta_{1,T-1}}\mathcal{Y}^T = \frac{\eta_{T-1}}{\theta_{1,T-1}^2}(\mathcal{W}^T + \mathcal{Y}^T),$$

thus

$$\max\{\mathbb{E}[f(w^T)], \mathbb{E}[f(y^T)]\} - f^\star$$

$$\leq \frac{\theta_{1,T-1}^2}{\eta_{T-1}}\mathbb{E}[\Psi^T]$$

$$\leq \frac{243\Delta}{(T-1+27(1+\omega))^2} \cdot \max\left\{\frac{9(1+\omega)^2(1+27(1+\omega))L}{T+27(1+\omega)}, \frac{200\omega(1+\omega)L}{3n}, 2L\right\}$$

$$= \mathcal{O}\left(\frac{(1+\omega^2/n)L\Delta}{T^2} + \frac{(1+\omega^3)L\Delta}{T^3}\right),$$

thus it suffices to achieve an $\epsilon$-solution with $\mathcal{O}\left(\left(1 + \frac{\omega}{\sqrt{n}}\right)\sqrt{\frac{L\Delta}{\epsilon}} + (1+\omega)\sqrt[3]{\frac{L\Delta}{\epsilon}}\right)$ iterations.  $\square$

## E    Correction on CANITA [33]

We observe that when $\omega \gg n$, the original convergence rate of CANITA [33] contradicts the lower bounds presented in our Theorem 2. This discrepancy may stem from errors in the derivation of equations (35) and (36) in [33], or from the omission of certain conditions such as $\omega = \Omega(n)$. To address this issue, we provide a corrected proof and the corresponding convergence rate. Here we modify the choice of $\beta_0$ in ([33], Theorem 2) to $9(1+b+\omega)^2/(2(1+b))$, while keeping all other choices consistent with the original proof, *i.e.*, $b = \min\{\omega, \sqrt{\omega(1+\omega)^2/n}\}$, $p_t \equiv 1/(1+b)$, $\alpha_t \equiv 1/(1+\omega)$, $\theta_t = 3(1+b)/(t+9(1+b+\omega))$, $\beta = 48\omega(1+\omega)(1+b+2(1+\omega))/(n(1+b)^2)$ and

$$\eta_t = \begin{cases} \frac{1}{L(\beta_0+3/2)}, & \text{for } t = 0, \\ \min\left\{\left(1 + \frac{1}{t+9(1+b+\omega)}\right)\eta_{t-1}, \frac{1}{L(\beta+3/2)}\right\}, & \text{for } t \geq 1. \end{cases}$$

By definition we have

$$\eta_T = \min\left\{\frac{T+1+9(1+b+\omega)}{1+9(1+b+\omega)}\eta_0, \frac{1}{L(\beta+3/2)}\right\}$$

$$= \min\left\{\frac{T+1+9(1+b+\omega)}{1+9(1+b+\omega)}\frac{1}{L(\beta_0+3/2)}, \frac{1}{L(\beta+3/2)}\right\}$$

$$\geq \min\left\{\frac{(T+9(1+b+\omega))(1+b)}{60L(1+b+\omega)^3}, \frac{1}{L(\beta+3/2)}\right\} \tag{40}$$

Plugging (40) and ([33],34) into ([33],33), we obtain

$$\mathbb{E}[F^{T+1}] = \mathcal{O}\left(\frac{(1+b+\omega)^3 L\Delta}{(T+9(1+b+\omega))^3} + \frac{(1+b)(\beta+3/2)L\Delta}{(T+9(1+b+\omega))^2}\right)$$

$$= \mathcal{O}\left(\frac{(1+b+\omega)^3 L\Delta}{T^3} + \frac{(1+b)(\beta+3/2)L\Delta}{T^2}\right). \tag{41}$$

Using $b = \min\{\omega, \sqrt{\omega(1+\omega)^2/n}\}$, we have

$$(1 + b + \omega)^3 = \Theta\left((1+\omega)^3\right),$$

and

$$(1+b)(\beta + 3/2) = \Theta\left((1+b) + \frac{\omega(1+\omega)(1+b+\omega)}{n(1+b)}\right)$$

$$= \Theta\left(1 + \frac{\omega^{3/2}}{n^{1/2}} + \frac{\omega^2}{n}\right),$$

thus (41) can be simplified as

$$\mathbb{E}[F^{T+1}] = \mathcal{O}\left(\frac{(1+\omega)^3 L\Delta}{T^3} + \frac{(1 + \omega^{3/2}/n^{1/2} + \omega^2/n)L\Delta}{T^2}\right).$$

Consequently, for $\epsilon < L\Delta/2$ (*i.e.*, a precision that the initial point does not satisfy), the communication rounds to achieve precision $\epsilon$ is given by $\mathcal{O}\left(\omega\frac{\sqrt[3]{L\Delta}}{\sqrt[3]{\epsilon}} + \left(1 + \frac{\omega^{3/4}}{n^{1/4}} + \frac{\omega}{\sqrt{n}}\right)\frac{\sqrt{L\Delta}}{\sqrt{\epsilon}}\right)$.

## F   Experimental details and additional results

This section provides more details of the experiments listed in Sec. 6, as well as a few new experiments to validate our theories.

### F.1   Experimental details

This section offers a comprehensive and detailed description of the experiments listed in Sec. 6, including problem formulation, data generation, cost calculation, and algorithm implementation.

**Least squares.** The local objective function of node $i$ is defined as $f_i(x) := \frac{1}{2}\|A_i x - b_i\|^2$, where $A_i \in \mathbb{R}^{M \times d}$, $b_i \in \mathbb{R}^M$. We set $d = 20$, $M = 25$, and the number of nodes $n = 400$. To generate $A_i$'s, we first randomly generate a Gaussian matrix $G \in \mathbb{R}^{nM \times d}$; we then apply the SVD decomposition $G = U\Sigma V^\top$ and replace the singular values in $\Sigma$ by an arithmetic sequence starting from 1 and ending at 100 to get $\tilde{\Sigma}$ and the resulted data matrix $\tilde{G} = U\tilde{\Sigma}V^\top$; we finally allocate the submatrix of $\tilde{G}$ composed of the $((i-1)M+1)$-th row to the $(iM)$-th row to be $A_i$ for all $1 \le i \le n$.

**Logistic regression.** The local objective function of node $i$ is defined as $f_i(x) := \frac{1}{M}\sum_{m=1}^M \ln(1 + \exp(-b_{i,m}a_{i,m}^\top x)$, where number of nodes $n = 400$, $a_{i,m}$ stands for the feature of the $m$-th datapoint in the node $i$'s dataset, and $b_{i,m}$ stands for the corresponding label. In a9a dataset, node $i$ owns the $(81(i-1)+1)$-th to the $(81i)$-th datapoint with feature dimension $d = 123$. In w8a dataset, node $i$ owns the $(120(i-1)+1)$-th to the $(120i)$-th datapoint with feature dimension $d = 300$.

**Constructed problem.** The local objective function of node $i$ is defined as

$$f_i(x) := \begin{cases} \frac{\mu}{2}\|x\|^2 + \frac{L-\mu}{4}([x]_1^2 + \sum_{1 \le r \le d/2-1}([x]_{2r} - [x]_{2r+1})^2 + [x]_d^2 - 2[x]_1), & \text{if } i \le n/2, \\ \frac{\mu}{2}\|x\|^2 + \frac{L-\mu}{4}(\sum_{1 \le r \le d/2}([x]_{2r-1} - [x]_{2r})^2), & \text{if } i > n/2, \end{cases}$$

where $[x]_l$ denotes the $l$-th entry of vector $x \in \mathbb{R}^d$. We set $\mu = 1$, $L = 10^4$, $d = 20$ and number of nodes $n = 400$.

**Compressors.** We apply various compressors to the algorithms with communication compression through our experiments. In the constructed quadratic problem, we consider ADIANA algorithm with random-$s$ compressors (see Example 1 in Appendix A) in six different settings, *i.e.*, three choices of $s$ ($s = 1, 2, 4$), with two different (shared or independent) randomness settings. In the least squares and logistic regression problems, we apply the independent random-$\lfloor d/20 \rfloor$ compressor to ADIANA, CANITA and DIANA algorithm. In particular, we use the unscaled version of the independent random-$\lfloor d/20 \rfloor$ compressor for EF21 to guarantee convergence, where the values of selected entries are transmitted directly to the server without being scaled by $d/s$ times. In Appendix F.2, we further apply independent *natural compression* [20] and *random quantization* [3] with $s = \lceil \sqrt{d} \rceil$ in the above algorithms.

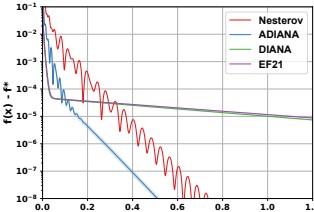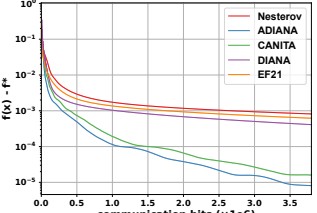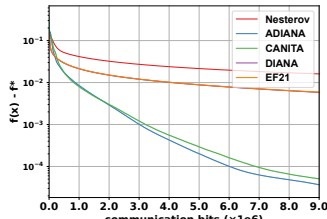

Figure 3: Convergence results of various distributed algorithms on a synthetic least squares problem (left), logistic regression problems with dataset `a9a` (middle) and `w8a` (right). The $y$-axis represents $f(\hat{x}) - f^\star$ and the $x$-axis indicates the total communicated bits sent by per worker. All compressors used are independent natural compression.

**Total communicated bits.** For non-compression algorithms and algorithms with a fixed-length compressor, such as random-$s$ and natural compression, the total communication bits can be calculated using the following formula: *total communication bits = number of iterations × communication rounds per iteration × communicated bits per round.* Among the algorithms we compare, ADIANA and CANITA communicate twice per iteration, while the other algorithms communicate only once. The communicated bits per round for non-compression algorithms amount to $64d$ for $d$ `float64` entries. In the case of the random-$s$ compressor, the communicated bits per communication are calculated as $64s + \lceil \log_2 \binom{d}{s} \rceil$. Similarly, for natural compression, the communicated bits per round are fixed at $12d$, with 1 sign bit and 11 exponential bits allocated for each entry. In the case of adaptive-length random quantization, the communication cost is evaluated using *Elias* integer encoding [16]. This cost is then averaged among $n$ nodes, providing a more representative estimate.

**Algorithm implementation.** We implement ADIANA, CANITA, DIANA, EF21 algorithms following the formulation in Algorithm 1, [33], [26], and [49], respectively. We implement Nesterov's accelerated algorithm with the following recursions:

$$\begin{cases} y^k = (1 - \theta_t)x^k + \theta_t z^k, \\ x^{k+1} = y^k - \eta_k \nabla f(y^k), \\ z^{k+1} = x^k + \frac{1}{\theta_k}(x^{k+1} - x^k). \end{cases}$$

The value of $\alpha$ in ADIANA, CANITA and DIANA are all set to $1/(1 + \omega)$, and we set $\gamma_k, \beta$ of ADIANA as in Theorem 3. Other parameters are all selected through running Bayesian Optimization [45] for the first $20\%$ iterations with 5 initial points and 20 trials. The exact value of the selected parameters are listed in Appendix F.3. Each curve (except for Nesterov's accelerated algorithm which does not involve randomness) is averaged through 20 trials, with the range of standard deviation depicted.

**Computational resource.** All experiments are run on an NVIDIA A100 server. Each trial consumes up to 10 minutes of running time.

### F.2 Additional experiments

**Addtional compressors.** In addition to the experiments in Sec. 6, we consider applying different compressors in the algorithms with communication compression. Fig. 3 and Fig. 4 show results of using natural compression and random quantization, respectively. These results are consistent with the results in Sec. 6.

**CIFAR-10 dataset.** We also consider binominal logistic regression with CIFAR-10 dataset, where labels of each datum are categorized by whether they equal to 3, *i.e.*, the corresponding figures belong to the `cat` category. The full training set with $50000$ images, are devided equally to $n = 250$ nodes. The compressor choices follow the same strategies as in Appendix F.1, where dimension $d = 3072$. Fig. 5 compares convergence results between Nesterov method and ADIANA with different compressors. It can be observed that ADIANA equipped with more aggressive compressors, *i.e.*, those with bigger $\omega$, benefits more from the compression, which is consistent with our theoretical results.

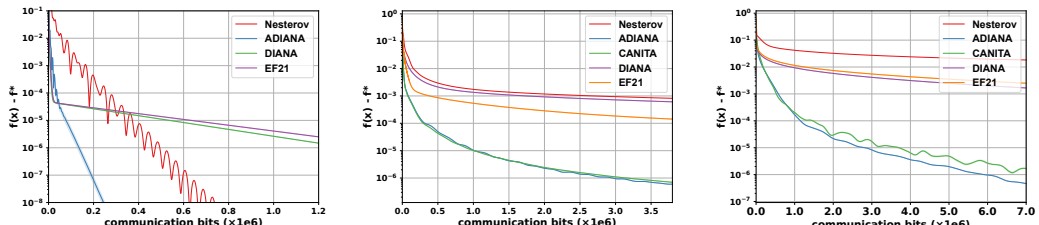

Figure 4: Convergence results of various distributed algorithms on a synthetic least squares problem (left), logistic regression problems with dataset a9a (middle) and w8a (right). The $y$-axis represents $f(\hat{x}) - f^\star$ and the $x$-axis indicates the total communicated bits sent by per worker. All compressors used are independent random quantization.

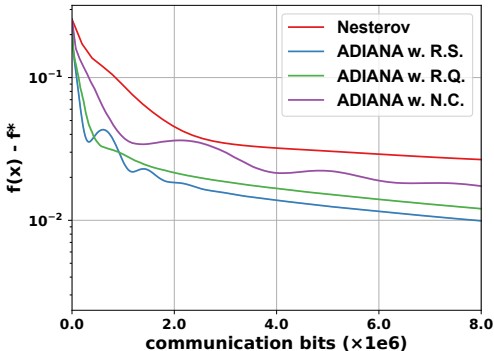

Figure 5: Experimental results of logistic regression problem on the CIFAR-10 dataset. The objective function is constructed by relabeling the 10 classes into 2 classes, namely cat (corresponding to the original cat class) and non-cat (corresponding to the rest classes). ADIANA w. R.S. / R.Q. / N.C. represents ADIANA algorithm with random-$\lfloor d/20 \rfloor$ compressor / random quantization compressor with $s = \lceil \sqrt{d} \rceil$ / natural compression compressor, where $d = 3072$ is the dimension of gradient vectors as well as the number of features in CIFAR-10 dataset. The experiments are conducted under the same setting as in the "Algorithm implementation" part in Appendix F.1.

### F.3 Parameter values

In this subsection, we list all the parameter values that are selected by applying Bayesian Optimization. Table 2, 3, 4, 5, 6 list the parameters chosen in the least squares problem, logistic regression using a9a dataset, logistic regression using w8a dataset, the constructed problem, and logistic regression using CIFAR-10 dataset, respectively.

Table 2: Parameters for algorithms in the least squares problem. Notation R.S. stands for independent random sparsification, N.C. stands for independent natural compression, R.Q. stands for independent random quantization.

| Algorithm | Parameters |
|---|---|
| Nesterov | $\eta = 3.0 \times 10^{-2}, \theta = 1.4 \times 10^{-2}.$ |
| ADIANA R.S. | $\eta = 4.8 \times 10^{-2}, \theta_1 = 2.2 \times 10^{-2}, \theta_2 = 7.6 \times 10^{-2}, p = 4.1 \times 10^{-2}.$ |
| ADIANA N.C. | $\eta = 3.9 \times 10^{-2}, \theta_1 = 1.0 \times 10^{-2}, \theta_2 = 2.9 \times 10^{-1}, p = 9.9 \times 10^{-1}.$ |
| ADIANA R.Q. | $\eta = 6.5 \times 10^{-2}, \theta_1 = 1.4 \times 10^{-2}, \theta_2 = 2.7 \times 10^{-1}, p = 5.5 \times 10^{-1}.$ |
| DIANA R.S. | $\gamma = 7.9 \times 10^{-2}.$ |
| DIANA N.C. | $\gamma = 7.4 \times 10^{-2}.$ |
| DIANA R.Q. | $\gamma = 7.6 \times 10^{-2}.$ |
| EF21 R.S. | $\gamma = 6.2 \times 10^{-2}.$ |
| EF21 N.C. | $\gamma = 6.8 \times 10^{-2}.$ |
| EF21 R.Q. | $\gamma = 7.4 \times 10^{-2}.$ |

Table 3: Parameters for algorithms in the logistic regression problem with a9a dataset. Notation $k$ stands for the index of iteration. Other notations are as in Table 2.

| Algorithm | Parameters |
|---|---|
| Nesterov | $\eta = 9.4 \times 10^{-1}, \theta = 1.7 \times 10^{-1}.$ |
| ADIANA R.S. | $\eta = 2.1, \theta_1 = \frac{1.3 \times 10^1}{k + 5.2 \times 10^2}, \theta_2 = 2.1 \times 10^{-1}, p = 7.7 \times 10^{-1}.$ |
| ADIANA N.C. | $\eta = 2.1, \theta_1 = \frac{1.0}{k + 4.3}, \theta_2 = 8.0 \times 10^{-3}, p = 8.0 \times 10^{-1}.$ |
| ADIANA R.Q. | $\eta = 2.2, \theta_1 = \frac{1.3}{k + 1.3}, \theta_2 = 1.5 \times 10^{-1}, p = 8.5 \times 10^{-1}.$ |
| CANITA R.S. | $\eta = \min\{\frac{k + 2.1 \times 10^2}{2.1 \times 10^2}, 1.4\}, \theta = \frac{2.0 \times 10^1}{k + 2.3 \times 10^2}, p = 7.8 \times 10^{-1}.$ |
| CANITA N.C. | $\eta = 1.2, \theta = \frac{2.1}{k + 1.2 \times 10^1}, p = 5.2 \times 10^{-1}.$ |
| CANITA R.Q. | $\eta = 2.0, \theta = \frac{3.0}{k + 3.0}, p = 7.2 \times 10^{-1}.$ |
| DIANA N.C. | $\gamma = 2.6.$ |
| DIANA R.S. | $\gamma = 9.4 \times 10^{-1}.$ |
| DIANA R.Q. | $\gamma = 4.7 \times 10^{-1}$ |
| EF21 R.S. | $\gamma = 1.3.$ |
| EF21 N.C. | $\gamma = 1.6.$ |
| EF21 R.Q. | $\gamma = 2.7.$ |

Table 4: Parameters for algorithms in logistic regression with `w8a` dataset. Notations are as in Table 3.

| Algorithm | Parameters |
|-----------|------------|
| Nesterov | $\eta = 1.5 \times 10^1, \theta = 9.4 \times 10^{-1}.$ |
| ADIANA R.S. | $\eta = \min\{\frac{k+4.1\times10^2}{1.2\times10^2}, 15\}, \theta_1 = \frac{8.8}{k+4.8\times10^2}, \theta_2 = 2.4 \times 10^{-2}, p = 3.6 \times 10^{-1}.$ |
| ADIANA N.C. | $\eta = 1.5 \times 10^1, \theta_1 = \frac{2.5}{k+1.1\times10^1}, \theta_2 = 6.7 \times 10^{-1}, p = 8.3 \times 10^{-1}.$ |
| ADIANA R.Q. | $\eta = 1.5 \times 10^1, \theta_1 = \frac{1.9}{k+7.4}, \theta_2 = 4.2 \times 10^{-1}, p = 9.9 \times 10^{-1}.$ |
| CANITA R.S. | $\eta = \min\{\frac{k+2.0\times10^2}{2.2\times10^2}, 7.7\}, \theta = \frac{1.1\times10^1}{k+2.3\times10^2}, p = 4.3 \times 10^{-1}.$ |
| CANITA N.C. | $\eta = \min\{\frac{k+1.1\times10^1}{2.7}, 1.1 \times 10^1\}, \theta = \frac{5.4}{k+7.4\times10^1}, p = 4.9 \times 10^1.$ |
| CANITA R.Q. | $\eta = \min\{\frac{k+1.6\times10^1}{7.3}, 1.5 \times 10^1\}, \theta = \frac{2.2}{k+2.2\times10^1}, p = 4.6 \times 10^{-1}.$ |
| DIANA R.S. | $\gamma = 1.5 \times 10^1.$ |
| DIANA N.C. | $\gamma = 1.6 \times 10^1.$ |
| DIANA R.Q. | $\gamma = 1.5 \times 10^1.$ |
| EF21 R.S. | $\gamma = 2.0 \times 10^1.$ |
| EF21 N.C. | $\gamma = 1.5 \times 10^1.$ |
| EF21 R.Q. | $\gamma = 1.5 \times 10^1.$ |

Table 5: Parameters for algorithms in the constructed problem. Notation i.d.rand-$s$ denotes independent random-$s$ compressor, s.d.rand-$s$ denotes random-$s$ compressor with shared randomness.

| Algorithm | Parameters |
|-----------|------------|
| Nesterov | $\eta = 1.4 \times 10^{-1}, \theta = 1.2 \times 10^{-4}.$ |
| ADIANA i.d.rand-1 | $\eta = 1.5 \times 10^{-4}, \theta_1 = 1.8 \times 10^{-1}, \theta_2 = 1.3 \times 10^{-1}, p = 1.5 \times 10^{-1}.$ |
| ADIANA i.d.rand-2 | $\eta = 1.5 \times 10^{-4}, \theta_1 = 1.5 \times 10^{-4}, \theta_2 = 5.0 \times 10^{-2}, p = 1.9 \times 10^{-1}.$ |
| ADIANA i.d.rand-4 | $\eta = 1.3 \times 10^{-4}, \theta_1 = 9.2 \times 10^{-2}, \theta_2 = 5.0 \times 10^{-2}, p = 2.3 \times 10^{-1}.$ |
| ADIANA s.d.rand-1 | $\eta = 1.4 \times 10^{-6}, \theta_1 = 2.0 \times 10^{-2}, \theta_2 = 1.6 \times 10^{-1}, p = 2.7 \times 10^{-2}.$ |
| ADIANA s.d.rand-2 | $\eta = 9.6 \times 10^{-6}, \theta_1 = 7.0 \times 10^{-2}, \theta_2 = 4.3 \times 10^{-1}, p = 1.8 \times 10^{-1}.$ |
| ADIANA s.d.rand-4 | $\eta = 1.6 \times 10^{-5}, \theta_1 = 6.0 \times 10^{-2}, \theta_2 = 2.1 \times 10^{-1}, p = 1.6 \times 10^{-1}.$ |

Table 6: Parameters for algorithms in logistic regression with CIFAR-10 dataset. Notations are as in Table 4.

| Algorithm | Parameters |
|-----------|------------|
| Nesterov | $\eta = 1.1 \times 10^{-1}, \theta = 1.5 \times 10^{-1}.$ |
| ADIANA R.S. | $\eta = 1.4 \times 10^{-1}, \theta_1 = \frac{12}{k+3.3\times10^2}, \theta_2 = 9.0 \times 10^{-2}, p = 4.3 \times 10^{-1}.$ |
| ADIANA N.C. | $\eta = 1.4 \times 10^{-1}, \theta_1 = \frac{1.0\times10^{-2}}{k+7.0}, \theta_2 = 7.0 \times 10^{-1}, p = 8.5 \times 10^{-1}.$ |
| ADIANA R.Q. | $\eta = 1.2 \times 10^1, \theta_1 = \frac{8.2}{k+59}, \theta_2 = 8.0 \times 10^{-1}, p = 6.0 \times 10^{-1}.$ |

