# OpenReview forum: "Unbiased Compression Saves Communication in Distributed Optimization: When and How Much?"
_NeurIPS.cc/2023/Conference — NeurIPS 2023 poster_

### Official Review · Reviewer_yR98 · 2023-06-16

**Soundness:** 3 good
**Presentation:** 3 good
**Contribution:** 3 good
**Rating:** 7
**Confidence:** 4

**Summary:**

The authors consider the distributed convex optimization problem in the centralized setting. In this setting, the authors provide the new lower bounds on the total communication cost under the assumption that nodes send unbiased independent compressed vectors to a master. They also provide improved analysis of the current state-of-the-art ADIANA method in the general convex and strongly convex regimes.

**Strengths:**

The paper provides two important things to the distributed optimization community:
1. While the upper bounds obtained by the ADIANA and CANITA methods are well known, finally, the authors provide lower bounds in the described setting.
2. The improved analysis of ADIANA is very surprising. I checked the proof. I'm not 100% sure, but the improved analysis seems to be correct. I believe that it is an interesting contribution on its own.
All in all, I think that this paper deserves to be published in NeurIPS.

**Weaknesses:**

However, I would like to point out some very important weakness that I want the authors to fix.

The authors compare the total communication cost of compressed methods with the accelerated (Nesterov's) method. They claim that it is possible to improve the complexity by $\min [n, \kappa].$ I agree with this fact but only under the assumption that **the local smoothness constants are equal to the global smoothness constant.** The author ignore the fact (or I missed it in the text, then sorry me for that) that the local smoothness constants can be $n$ times larger than the global one. The authors define the local $L$--smoothness constant as
$$f_i(y) \leq f_i(x) + <\nabla f_i(x), y - x> + \frac{L}{2} ||x - y||^2 \quad \forall i$$ While the non-compressed methods define it as
$$f(y) \leq f(x) + <\nabla f(x), y - x> + \frac{L}{2} ||x - y||^2.$$ The $L$--constant of $f_i$  can be $n$ times larger than $L$--constant of $f$! I want to ask the authors to clarify it clearly in the paper. Ideally, in the abstract and in the contributions.

**Questions:**

-

**Limitations:**

-

---

> ### Author Rebuttal · Authors · 2023-08-02
>
> We thank the reviewer for the positive comments and the valuable question.
>
> As the reviewer accurately points out, the convergence analysis presented in our paper is contingent upon the condition on local smoothness constants. Under the difference of local and global smoothness, it remains uncertain whether our assertion regarding the potential communication cost reduction by independent unbiased compressors, on the order of $\Theta(\sqrt{\min\\{n,\kappa\\}})$, remains valid.
>
> Despite our earnest efforts, we have been unable to clarify this open question within the confines of the demanding rebuttal deadline. Consequently, to address the reviewer's valid concern, we will make revisions in our abstract and contribution sections and explicitly clarify that our claims hold under the assumption that the local smoothness and the global smoothness constants are constrained by a common upper bound $L$.  For example, we will revise statements in Line 18-19 in the abstract as "Our results reveal that using independent unbiased compression can reduce the total communication cost by a factor of up to $\Theta(\sqrt{\min\\{n,\kappa\\}})$ under the assumption that all local smoothness constants are constrained by a common upper bound $L$".
>
> Additionally, we will also revise statements in Line 100-101 in the introduction as "independent unbiased compression can decrease total communication costs by up to $\Theta(\sqrt{\min\\{n,\kappa\\}})$ when all local smoothness constants  are constrained by a common upper bound $L$."
>
> On the other hand, we hope the reviewer can understand that **it is not uncommon to make the assumption that all local functions share a common upper bound of smoothness  in literature on distributed optimization**, which can significantly simplify the convergence analysis. For example, most algorithms listed in Table 1 in our paper such as CGD [20], ACGD [25], DIANA [32],  ADIANA [25], CANITA [26], and NEOLITHIC [13] assumes $L$-smoothness across all local functions.
>
> We hope this response can resolve the reviewer's concerns. We are looking forward to the follow-up discussion with the reviewer, and more than happy to address any further comments or questions.

---

> > ### Comment · Reviewer_yR98 · 2023-08-10
> > **Respond**
> >
> > Thank you for the rebuttal!
> >
> > I didn't mean that you would start reproving everything for the "global smoothness" case. I only kindly asked you to add comments in the paper that the comparison between your results and the classical AGD method are only valid under the assumption that "that all local smoothness constants are constrained by a common upper bound." From the rebuttal, it is clear that we are on the same page!
> >
> > The authors addressed all my questions, and I will  keep my score.
> >
> > BTW, I've recently found the paper \[1\]. They consider a slightly different setup, bidirectional communication. However, in their Contribution   C, they say that they improve CANITA's $1 / \varepsilon^{1/3}$ rate to $\ln 1/ \varepsilon.$ In your paper, you have the gap between the upper bound and the lower bound ($1 / \varepsilon^{1/3}$ vs $\ln 1/ \varepsilon.$). It seems that this paper closes that gap, so your lower bound is tight.
> >
> > \[1\]: https://arxiv.org/pdf/2305.12379.pdf

---

> > > ### Author Response · Authors · 2023-08-11
> > >
> > > We really appreciate your follow-up feedback. We'll definitely clarify the smoothness issue in the revision as we stated in our rebuttal.
> > >
> > > We are also happy to discuss the recent paper of 2Direction with the reviewer. As mentioned by their authors, the rate of 2Direction in the low accuracy regimes improves the $\Theta\left(\frac{1}{\epsilon^{1/3}}\right)$ term to $\Theta\left(\log\frac{1}{\epsilon}\right)$. However, when simplfiying to our setting where $r=0$, $L=L_{\max}$, $\alpha=1$, their dominant term in the high precision case (namely, $\epsilon$ is sufficiently small) $\left(1+\frac{\omega^{1/2}}{n^{1/6}}+\frac{\omega^{3/4}}{n^{1/4}}+\frac{\omega}{\sqrt{n}}\right)\frac{\sqrt{L\Delta}}{\sqrt{\epsilon}}$ is worse than  $\left(1+\frac{\omega}{\sqrt{n}}\right)\epsilon^{-1/2}$, the major term in our lower bound. Therefore, despite of the inspiring work, it remains open to find an algorithm that, in the generally-convex case, can tightly match both the $\ln(1/\epsilon)$ term and the $1/\sqrt{\epsilon}$ term simultaneously.
> > >
> > > Finally, we thank the reviewer for pointing out this recent work (which came out after our submission) and we will also comment on this work in our later revisions.

---

> > > > ### Comment · Reviewer_yR98 · 2023-08-11
> > > > **Respond**
> > > >
> > > > I'm in no way trying to down the value of your contribution. Everything that you wrote is almost true. Indeed, they do not improve the high precision term of the original CANITA paper.
> > > >
> > > > BTW, using Young's inequality, note that $\frac{\omega^{1/2}}{n^{1/6}} \leq \frac{1}{3} 1^{3} + \frac{2}{3} \frac{\omega^{3/4}}{n^{1/4}}.$ so their complexity is $\Theta(1 + \frac{\omega^{3/4}}{n^{1/4}} + \frac{\omega}{\sqrt{n}})\frac{\sqrt{L \Delta}}{\sqrt{\varepsilon}}$ when $L = L_{\max}$ in the high precision case.
> > > >
> > > > Good luck with other rebuttals!

---

### Official Review · Reviewer_JbF9 · 2023-06-28

**Soundness:** 2 fair
**Presentation:** 1 poor
**Contribution:** 2 fair
**Rating:** 3
**Confidence:** 2

**Summary:**

The authors study the extent to which unbiased compression can reduce the total communication cost of optimizing smooth convex functions. Additionally, they refine the analysis of ADIANA ([25]) to show that the bounds are near-optimal.

**Strengths:**

Quantifying how much unbiased compression helps is an interesting topic. While I believe that nearly anyone (that uses compression) works with unbiased compressors in the distributed case, it is interesting to learn by how much it helps compared with biased compressors.



**Weaknesses:**

It is not very surprising to me what without independence, unbiasedness cannot help in general.
Further, I do not recall seeing works that do not use independence to get the error-cancellation, *other than works that use correlated compression to drive the error lower than that of i.i.d. compressors* (e.g.,  Correlated quantization for distributed mean estimation and optimization, ICML 2022).

Also, the writing is hard to follow and seems to be contradictory in places.
For example, You write, "Compared to lower bounds when using unbiased compression without independence [13], our lower bounds demonstrate significant improvements when n and κ are large".
However, the upper and lower bounds of [13] seem to match in your Table 1, while your lower bound seems *weaker* than that of [13].

In general, there cannot be two matching sets of upper and lower bounds.

Some of the claims also seem inflated; I don't view $O(\omega \epsilon^{-O(1)})$ as being near optimal for a lower bound of $\Omega(\omega\log\epsilon^{-1})$ -- the dependency on $\epsilon$ is exponentially worse.

The improvement in the analysis of ADIANA is only applicable to some parameter range as it shaves an additive factor.

(As a side note, it took me a minute to understand that by **ADIANA (Ours)** you don't actually mean that it's your *algorithm*, but just a refined analysis.)





**Questions:**

Can you please clarify the differences in the asymptotics with [13]? (Also, what is specified in the table seems to be different than in their paper, where they have an additional additive term.)



**Limitations:**

Seems fine

---

> ### Author Rebuttal · Authors · 2023-08-02
>
> We thank the reviewer for the detailed comments. We have clarified all questions as clearly as possible, and will be glad to address any further comments or questions.
>
> **1. Not surprising that unbiasedness cannot help without independence**
>
> We respectfully disagree with this opinion. Lots of literature believes that compression always saves the total communication costs, regardless of the relationship between compressors. While the reviewer's intuition is correct, it is essential to provide rigorous proofs, as what we contribute in the paper, to translate intuition into facts. It would be inappropriate to make definitive claims without in-depth mathematical clarifications.
>
> **2. Rare works do not use independence**
>
> Please see Point 3 in the "global response".
>
> **3. Contradictions in lower bounds**
>
> Please note that our lower bound/optimal complexity is proved in a different setup from [13]. Specifically, [13] does not assume the independence of compressors. In contrast, we impose the independence of compressors and only consider the lower bound and convergence of algorithms under this additional assumption. Due to the difference in setups, our results are different from and do not contradict with [13].
>
> Furthermore, as supported with our upper bound part, our lower bounds are nearly tight (at least in the strongly convex case) and thus represent the optimal complexities. Given the fact that ours and [13] are both optimal complexities in their respective setups (with or without independence), our complexities improves [13] by factors depending on $n$, $\kappa$, and $\omega$ (also see the response in point 7). This reveals the nontrivial advantage brought by independence of compressors. We hope this clarifies the reviewer's confusion.
>
> **4. Inflated claims**
>
> To resolve reviewer's concern, we will replace the statement of "nearly optimal" with "nearly optimal in the strongly convex case" throughout the paper to avoid inflated claims.
>
> The polynomial gap exists in the generally convex (GC) case whereas our result in the strongly convex (SC) case is polynomially tight. However, it is worth noting that even with such gap, our result is still the **state-of-the-art** in the GC case.  The existence of this tiny gap does not affect the major conclusion in our paper: independent unbiased compression saves communication costs up to $\sqrt{\min\\{n,\kappa\\}}$ times while unbiased compression alone cannot. The established result in the GC case still outperforms the complexity in [13] with advantage at most $\sqrt{\min\\{n,\kappa\\}}$ times when $\epsilon\lesssim (\frac{1+\omega/\sqrt{n}}{1+\omega})^6$ such that the $\epsilon^{-1/3}$ term is dominated.
>
> **5. ADIANA improvement**
>
> Our improvement over existing ADIANA analysis is significant.
>
> - First, the additive term $\omega^{3/4}/n^{1/4}\sqrt{\kappa}$ shaved by us is not trivial and can **dominate** in certain regimes. When $\min\\{n,\kappa^2/n\\}\gtrsim \omega\gtrsim n^{1/3}$, the term $\omega^{3/4}/n^{1/4}\sqrt{\kappa}$ dominates $\sqrt{\kappa}$, $\omega\sqrt{\kappa} /\sqrt{n}$, and $\omega$, and thus the rate of ADAINA in [25] becomes $\omega^{3/4}/n^{1/4}\sqrt{\kappa}\ln(1/\epsilon)$. Compared to our complexity in this case, the ratio is
> $$\frac{\mathrm{ADIANA}[25]}{ \mathrm{ADIANA [Thm3]}} \asymp \frac{ \omega^{3/4}/n^{1/4}\sqrt{\kappa} }{ \\omega+(1+\\omega/\sqrt{n} )\sqrt{\kappa} }\asymp \min\left\\{\frac{\sqrt{\kappa}}{(n\omega)^{1/4}}, \frac{\omega^{3/4}}{n^{1/4}},\frac{n^{1/4}}{\omega^{1/4}}\right\\}\gtrsim 1.$$
> Here $\lesssim$, $\gtrsim$, and $\asymp$ denote (in)equalities that hold up to a numeric constant.
> Note that the ratio can be as large as $\min\\{\kappa^{3/8}/n^{1/4} ,n^{1/8}\\}$ if compressors are with $\omega\asymp \min\\{\sqrt{n}, \sqrt{\kappa}\\}$. Therefore, when $\kappa \gtrsim n$ and $\omega\asymp \sqrt{n}$, our improved rate is faster than the original one by $n^{1/8}$ which is significant as $n$, the number of workers, can be large.
>
> - Second, shaving the additive term directly leads ADIANA to tightly matches with the lower bound in the SC case under independent unbiased compression. This milestone has not been attained or even explored in literature, to our knowledge.
>
> - Third, we establish the first convergence of ADIANA in the GC case, while exsiting ADIANA does not. In addition, our convergence rate outperforms all existing literature.
>
> **6. ADIANA (Ours)**
>
> Will rephrase as ADIANA (Thm3).
>
> **7. Asymptotic differences with [13]**
>
> First, [13] studies stochastic optimization where gradients are noisy with $\sigma^2$-bounded variances (see [13, Assumption 3]). Our work instead considers deterministic distributed optimization where gradients are noiseless, corresponding to the case of $\sigma =0$. The results of [13] can apply to noiseless gradients by setting $\sigma^2$-dependent terms to zero. This is the reason why we removed the $\sigma$-dependent terms in our table.
>
> We are not sure what the word "asymptotic" mean by the reviewer. We conjecture the reviewer indicates the simplified rates of [13] when the additional $\sigma$-dependent terms removed. In this case, the main difference of ours and [13] lies in the independence of compressors. In [13], compressors are not assumed to be independent and those complexities also apply to non-iid compressors. As we compared in Sec 5 in details, our rate is smaller than theirs justified by the factor
> $$\frac{(1+\omega)\sqrt{\kappa}}{\omega+(1+\omega/\sqrt{n})\sqrt{\kappa}}=\frac{1}{1+\omega}+\frac{\omega}{1+\omega}\left(\frac{1}{\sqrt{n}}+\frac{1}{\sqrt{\kappa}}\right)\asymp\frac{1}{\min\\{1+\omega,\sqrt{n},\sqrt{\kappa}\\}}.$$
> When adopting compressors with $\omega \gtrsim\min\\{\sqrt{n},\sqrt{\kappa}\\}$, our rate is $\min\\{\sqrt{n},\sqrt{\kappa}\\}$ times faster. The improvement mainly comes from exploiting the independence across compressors while [13], though more generally applicable to correlated compressors, inevitably sacrifices convergence.

---

> > ### Author Response · Authors · 2023-08-20
> > **Can we have your response to our rebuttals?**
> >
> > Dear Reviewer JbF9,
> >
> > The reviewer-author discussion period will end **tomorrow**. Could you please let us know whether our rebuttal has resolved your concerns? If not, could you please point them out so that we can try to address them as best as we can?
> >
> > Thanks very much for your time and efforts to review our work.

---

> ### Author Response · Authors · 2023-08-14
> **Need more clarifications?**
>
> Dear reviewer JbF9,
>
> We thank you for your valuable comments. We have made detailed responses to address your concerns, but we have not received your replies to our current clarifications yet. We thus kindly ask if our responses have addressed all your concerns. If not, we are more than happy to provide more clarifications.
>
> Best,
>
> The authors of paper 5699

---

### Official Review · Reviewer_MRty · 2023-07-06

**Soundness:** 3 good
**Presentation:** 3 good
**Contribution:** 3 good
**Rating:** 7
**Confidence:** 4

**Summary:**

This paper investigates the communication saving of unbiased compression on distributed optimization. The main contributions are:

i) a communication lower bound of distributed optimization algorithms with (not necessarily independent) unbiased compression;

ii) a communication lower bound of distributed optimization algorithms with (independent) unbiased compression;

iii) refined analysis of ADIANA to show tightness of these bounds;

iv) discussions on the importance of independence to the communication saving.

**Strengths:**

i) Overall, this paper is well written and easy to follow.

ii) The investigated problem is important to distributed optimization, and the results are convincing.

**Weaknesses:**

i) Section 1 claims that quantization and sparsification are modeled as unbiased compressors in some papers. But they can be biased too. Please clarify.

ii) The error cancellation explanation of independence to unbiased compression makes sense. But, if $x_i$ at different nodes are quite different, do we still need independence to achieve the communication saving?

iii) Table I considers both convex and strongly convex cases. Are there any difficulties in analyzing the non-convex case?

iv) Some bounds in Table 1 are summations of two terms, for example, $\ln(1/\epsilon)$ and $\epsilon^{-1/2}$, $\epsilon^{-1/3}$ and $\epsilon^{-1/2}$, etc. Is it possible to reduce to one term?

v) Which compressors satisfy Assumption 2 (unbiased compressor) and Assumption 3 (independent compressor)? Please comment below the assumptions.

vi) The communication saving is discussed via comparing the lower bounds. However, in some datasets some algorithms might perform much better than the lower bounds. In this case, the comparisons would yield different conclusions. Please comment on this.

vii) I give a score of 6 at this stage, and would like to change it after the rebuttal period.

**Questions:**

i) Section 1 claims that quantization and sparsification are modeled as unbiased compressors in some papers. But they can be biased too. Please clarify.

ii) The error cancellation explanation of independence to unbiased compression makes sense. But, if $x_i$ at different nodes are quite different, do we still need independence to achieve the communication saving?

iii) Table I considers both convex and strongly convex cases. Are there any difficulties in analyzing the non-convex case?

iv) Some bounds in Table 1 are summations of two terms, for example, $\ln(1/\epsilon)$ and $\epsilon^{-1/2}$, $\epsilon^{-1/3}$ and $\epsilon^{-1/2}$, etc. Is it possible to reduce to one term?

v) Which compressors satisfy Assumption 2 (unbiased compressor) and Assumption 3 (independent compressor)? Please comment below the assumptions.

vi) The communication saving is discussed via comparing the lower bounds. However, in some datasets some algorithms might perform much better than the lower bounds. In this case, the comparisons would yield different conclusions. Please comment on this.

**Limitations:**

N/A.

---

> ### Author Rebuttal · Authors · 2023-08-02
>
> We thank the reviewer for the detailed comments. All questions have been clarified as best as we can. We are glad to address any further comments or questions.
>
> **1. Section 1 claims that quantization and sparsification are modeled as unbiased compressors in some papers. But they can be biased too.**
>
> Thanks for pointing it out. In Section 1 (Line 34~36) we wrote: *In literature [3, 20, 15], these compression techniques are often modeled as a random operator $C$, which satisfies the properties of unbiasedness $\mathbb{E}[C(x)] = x$ and $\omega$-bounded variance $\mathbb{E}[\\|C(x) − x\\|^2] \leq \omega\\|x\\|^2$.* We did not mean that all these compressors are all modeled as unbiased ones, and we'll address the existence of biased compressors in later revisions to prevent misleading.
>
> Specifically, we will add the following comments directly after Line 37:
>
> *Besides, part of the compressors can also be modeled as biased estimators with $\mathbb{E}[\\|C(x)-x\\|^2]\leq(1-\delta)\\|x\\|^2$ where $\delta\in(0,1]$ [13,39,40].*
>
> **2. If $x_i$ at different nodes are quite different, whether we still need independence to achieve the communication saving.**
>
> As long as the independence exists, the aggregation of compressed messages always benefits from the error cancellation because all the cross terms in the expansion of $\mathbb{E}[\\|\frac{1}{n}\sum_{i=1}^n C_i(x_i)-x_i\\|^2]$ are zero in expectation so that $\mathbb{E}[\\|\frac{1}{n}\sum_{i=1}^n C_i(x_i)-x_i\\|^2]=\frac{\omega}{n^2}\sum_{i=1}^n\\|x_i\\|^2$. As for non-independent compressors, in the worst case, there remains the possibility that the compression error of the aggregated messages approaches to the upper bound obtained from Cauchy's inequality, i.e., $\mathbb{E}[\\|\frac{1}{n}\sum_{i=1}^n C_i(x_i)-x_i\\|^2]\approx \frac{\omega}{n}\sum_{i=1}^n\\|x_i\\|^2$, which can be greater than the one of independent compressors, up to $n$ times.
>
> **3. Difficulties in analyzing the non-convex case.**
>
> Unfortunately, in the non-convex scenario, we have not found any algorithm that belongs to the considered family and is capable to achieve provable savings in total communication costs. One close work is  (MARINA: Faster Non-Convex Distributed Learning with Compression, ICML 2021) which exhibits an extraordinary convergence rate but requires transmitting full (i.e., uncompressed) gradients occasionally, making it jumping out of the algorithm class we consider. As for ADIANA, it is specially designed for the convex case, and it may not even converge in the non-convex case.
>
> **4. Whether summations like $\ln(1/\epsilon)$ and $\epsilon^{-1/2}$, $\epsilon^{-1/3}$ and $\epsilon^{-1/2}$, etc. are possible to be reduced to one term?**
>
> Each term can be the dominating one depending on the magnitude of precision $\epsilon$ and the setup-related constants including $L$ and $\Delta$. For example, in the lower bound of the generally-convex case:
> \begin{equation}
> \tilde{\Omega}\left(\omega\ln\left(\frac{1}{\epsilon}\right)+\left(1+\frac{\omega}{\sqrt{n}}\right)\frac{\sqrt{L\Delta}}{\sqrt{\epsilon}}\right),
> \end{equation}
> The $\ln(1/\epsilon)$ term can dominate when the parameters satisfy $\omega=\sqrt{n}\gg L\Delta/\epsilon$, and the $1/\sqrt{\epsilon}$ term can dominate in a high-precision regime where $\frac{\ln(1/\epsilon)}{1/\sqrt{\epsilon}}$ is sufficiently small.
>
> In the upper bound of CANITA in the generally-convex case:
> \begin{equation}
> \mathcal{O}\left(\omega\frac{\sqrt[3]{L\Delta}}{\sqrt[3]{\epsilon}}+\left(1+\frac{\omega^{3/4}}{n^{1/4}}+\frac{\omega}{\sqrt{n}}\right)\frac{\sqrt{L\Delta}}{\sqrt{\epsilon}}\right),
> \end{equation}
> The $1/\sqrt[3]{\epsilon}$ term can dominate when $\omega=n^{1/3}\gg \left(\frac{L\Delta}{\epsilon}\right)^{1/6}$, and the $1/\sqrt{\epsilon}$ term can dominate in a high-precision regime where $\epsilon$ is sufficiently small.
>
> Therefore, these summations should not be reduced into one term without further conditions.
>
> **5. Which compressors satisfy Assumption 2 (unbiased compressor) and Assumption 3 (independent compressor)?**
>
> We'll comment the following examples under Assumption 3:
>
> *In fact, lots of compressors in the literature satisfy Assumption 2, see, e.g., standard dithering [3, Lemma 3.1], natural compression [14, Thm 3], natural dithering [14, Thm 8], ternary quantization [48]. As long as the worker-associate compressors have irrelavant sources of randomness (e.g., using different seeds for randomness), they further satisfy Assumption 3.*
>
> We also refer the reviewer to Point 3 in the "global response" where we show examples that independent compressors can be transformed into dependent compressors when using the same random seed.
>
> **6. The communication saving is discussed via comparing the lower bounds. However, in some datasets some algorithms might perform much better than the lower bounds. In this case, the comparisons would yield different conclusions.**
>
> As described, our lower bounds consider the worst case of the objective function that satisfy the $L$-smoothness and convexity. Thus our lower bounds only provide insights and laid down the standard for comparing the theoretical convergence rate relying merely on the $L$-smoothness and convexity.
>
> We'll comment this under Theorem 1:
>
> *It is worth noting that the theoretical results can vary from practical observations due to the particularities of certain datasets, where these datasets may exhibit additional structures with which algorithms can enjoy faster convergence, compared to the one theoretically justified without using any additional condition. However, such studies are beyond the scope of our work and we will leave it for future work.*
>
>
>
> We thank the reviewer again for his careful and valuable comments. We hope these response can clarify the reviewer's questions. We are looking forward to the follow-up discussion with the reviewer, and more than happy to address any further comments or questions.

---

> > ### Comment · Reviewer_MRty · 2023-08-19
> >
> > I have read the response and increased my score to 7.

---

> > > ### Author Response · Authors · 2023-08-19
> > > **Thanks very much for raising the score**
> > >
> > > We are very happy that your concerns have been addressed. Thanks very much for your valuable feedback and comments.

---

> ### Author Response · Authors · 2023-08-14
> **Need more clarifications?**
>
> Dear reviewer MRty,
>
> We thank you for your valuable comments. We have made detailed responses to address your concerns, but we have not received your replies to our current clarifications yet. We thus kindly ask if our responses have addressed all your concerns. If not, we are more than happy to provide more clarifications.
>
> Best,
>
> The authors of paper 5699

---

### Official Review · Reviewer_u1Ro · 2023-07-18

**Soundness:** 3 good
**Presentation:** 3 good
**Contribution:** 3 good
**Rating:** 7
**Confidence:** 3

**Summary:**

The paper analyzes the communication cost of unbiased compression algorithms for distributed optimization and proves that unbiased compression alone cannot reduce communication cost since whatever is saved via compression is lost due to an increase in the number of communication rounds needed for convergence. The authors then show that if in addition to being unbiased if the compressors are also independent then the lower bound on the number of communication rounds is reduced by a factor of $\Theta(\sqrt(\min(n, \kappa)))$ where $n$ is the number of nodes and $\kappa$ is the condition number of the function being minimized. They then improve the convergence analysis of ADIANA, an existing algorithm for communication compresssion in distributed optimization, and show that the lower bounds can be matched upto a log factor. Simulations on distributed least squares and distributed logistic regression corroborate their analysis and validate their claims.

**Strengths:**

1. The authors identify an important gap in the existing literature on communication compression where emphasis is primarily laid on the unbiasedness of compressors, without also considering the effect of independence. They systematically build and explain their analysis providing theoretical proofs for all their claims. The theoretical analysis coupled with the clear explanation can make this an important and impactful addition to the literature.

2. Considering the total communication cost (per-round cost x number of rounds) makes the analysis more realistic and complete than previous works which just show savings in either per-round cost or on the number of rounds. Moreover, by proposing a general lower bound for all distributed optimization algorithms with unbiased and independent communication compression this work clarifies the target communication cost for all such communication compression algorithms which can make them easier to evaluate and analyze theoretically.

**Weaknesses:**

1. The writing is a bit heavy on notation and the analysis may be hard to follow for readers unfamiliar with the literature. Especially since their is not clear example or intuition provided in the main paper to indicate why independence may help.

2. It is a bit unclear why ADIANA is chosen for analysis. The transition from deriving a general lower bound to analyzing a specific algorithm feels a bit abrupt. If possible, it would be good to provide some intuition for the kind of properties that would make a communication compression algorithm likely to be close to the lower bound thereby justifying the choice of ADIANA for analysis. This is a minor point however, as I understand that such an intuition may not be available at this time so it is okay if that is the case.

**Questions:**

1. To address the first point under weaknesses above, I would recommend moving the content of Appendix A which provides intuition on why independence helps to the main pape. Algorithm 1 can be moved to the Appendix to make space, since it appears to be just a reproduction of the ADIANA algorithm from [25].

2. Can you also provide an example of a compression scheme where the compressors are unbiased but not mutually independent and provide some intution for its sub-optimality or show it through any of the experiments in Section 6?

3. Were any changes made to the ADIANA algorithm at all in your analysis or experiments or was the algorithm the same as that in [25] everywhere?

**Limitations:**

The authors have identified the major limitation of the analysis being limited to convex functions only and have marked that as an issue to be addressed in future work which is fine with me. I have identified a couple of more minor limitations under Weaknesses above and look forward to the authors' responses to those.

---

> ### Author Rebuttal · Authors · 2023-08-02
>
> Thank you for your positive feedback and valuable comments. Below are our answers to the concerns you proposed.
>
> **1. I would recommend moving the content of Appendix A which provides intuition on why independence helps to the main pape. Algorithm 1 can be moved to the Appendix to make space, since it appears to be just a reproduction of the ADIANA algorithm from [25].**
>
> Thanks for the great suggestions! We put the argument of intuition in Appendix mainly because of the page limit. We will re-organize the paper as the reviewer suggested in the revision.
>
> **2. Can you also provide an example of a compression scheme where the compressors are unbiased but not mutually independent and provide some intuition for its sub-optimality or show it through any of the experiments in Section 6?**
>
> - A proper example can be the scaled random sparsification where each worker randomly chooses $s$ entries out of the total $d$ entries, scales their values to guarantee unbiasedness, and transimts them to the server, see Appendix B in our paper for reference. When each worker samples $s$ entries randomly and independently, the compressors are mutually indepedent and unbiased. When all workers **share the same random seed** to sample $s$ entries, the compressors are unbiased but correlated with each other.
>
> - According to the above example, it is evident that if compressors are independent, then the indices of transmitted entries by all workers are likely to be uniformly diverse, and the server apparently can observe more entries in this case and thus the compression-incurred information distortion is milder. In contrast, when using the same random seed, all compressors will sample entries with the same indices, and the server will observe much fewer entries in this scenario. This would explain the intuition behind the sub-optimality of mutually dependent unbiased compressors.
>
> - The sub-optimality of the dependent case can be viewed through Line 342~347 in Section 6, where we can see clearly through Figure 1 that ADIANA with random-$s$ compressors with shared randomness behave much worse than those with identical independent compressors.
>
> - Unbiased compressors with the same seed are still widely used in practice due to its compatibility with All-Reduce operation, a highly effective protocol of distributed gradient aggregation used by default in PyTorch and TensorFlow, see Point 3 in the "global response" for more details. For example, unbiased compressors with the same seed are used in references [R1] and [R2] listed below.
>
> [R1] T. Vogels et. al., PowerSGD: Practical Low-Rank Gradient Compression for Distributed Optimization, arXiv:1905.13727
>
> [R2] C. Xie et. al., CSER: Communication-efficient SGD with Error Reset, arXiv:2007.13221.
>
> **3. Were any changes made to the ADIANA algorithm at all in your analysis or experiments or was the algorithm the same as that in [25] everywhere?**
>
> At the level of algorithmic design, the only difference in ours is to remove the proximal mapping as we consider the smooth case. However, **the improvement of analysis is significant**. We adopt effective choices of parameters (such as $\theta$, $\alpha$, $\beta$) to obtain an improved rate and thus attain the optimality in the strongly convex case. Furthermore, the analysis in the generally convex case is brand new and no convergence result of ADIANA exists in the generally convex case, to our best knowledge.
>
> **4. Justfying the choice of ADIANA to achieve lower bound**
>
> We choose to improve upon ADIANA mainly because **its previous convergence rate (in the strongly convex case) is the closest to our lower bound among all existing algorithms**. Another intuition is that DIANA-type algorithms is known to **benefit from independence of compressors** in the sense that it provably outperforms the best achievable rate proved by [13] when using non-independent compressors. While we believe there can be other algorithms with similar performance, we simply choose to improve upon ADIANA to verify the tightness of our lower bounds, which is important in addressing the *how much* question we concern.
>
> We thank the reviewer again for his careful and valuable comments. We hope these response can clarify the reviewer's questions. We are looking forward to the follow-up discussion with the reviewer, and more than happy to address any further comments or questions.

---

> > ### Comment · Reviewer_u1Ro · 2023-08-15
> > **Re**
> >
> > Thank you for the detailed response. The example and intuition provided for unbiased but dependent compressors makes sense to me and I think adding these points to the paper would definitely help readers understand why incorporating independence can lead to tighter bounds and lower communication cost in practice. I do not have any other questions or concerns and as I have already recommended acceptance, I will keep my score.

---

> > > ### Author Response · Authors · 2023-08-19
> > > **Thanks very much for the follow-up comments**
> > >
> > > We are very happy that your concerns have been addressed. Following your suggestions, we will add these examples and intuitions to the main paper in the revision. Thanks very much for your valuable feedback and comments.

---

### Official Review · Reviewer_tEWM · 2023-07-26

**Soundness:** 4 excellent
**Presentation:** 3 good
**Contribution:** 3 good
**Rating:** 7
**Confidence:** 1

**Summary:**

Several communication compression strategies have been proposed in the last few years to improve distributed optimization. The authors consider the tradeoffs between lowering the communication costs per round vs the number of communication rounds and understand how they affect total communication costs. They theoretically formulate this and prove that under some assumptions, unbiased compression might not save total communication. However,  if the compressors used by all workers are further assumed independent, they prove that total communication cost can be decreased. They also prove lower bounds on certain classes of algorithms by refining the analysis for ADIANA.

**Strengths:**

The results are provably improved. The presentation is clear and easy to follow.

**Weaknesses:**

The lower bounds are proved only for certain classes of algorithms.

**Questions:**

It would be better if some experiments from the appendix were moved to main body.

---

> ### Author Rebuttal · Authors · 2023-08-02
>
> We thank the reviewer for the positive comments. All questions have been clarified below.
>
> **1. The lower bounds are proved only for certain classes of algorithms.**
>
> We agree that our lower bounds only apply to the class of certain algorithms described in Section 2.3. However, these settings are broad enough to cover most first-order algorithms in literature, e.g., [38, 25, 20, 61] , as well as all algorithms listed in Table 1. Therefore, our established lower bounds remain valuable and sufficient to address questions Q1 and Q2 raised in the introduction.
>
> **2. It would be better if some experiments from the appendix were moved to main body.**
>
> We temporarily put these experiments into Appendix because of the page limit. We will reorganize them once we have more space allowed. Specifically, we will move Figure 4 and 5 to Section 6 next to Figure 2 and modify the description in Line 339~340 to:
>
> *We set $n=400$ and separately choose independent random-$\lfloor d/20\rfloor$ compressors (Figure 2), independent natural compression (Figure 4) and independent random quantization compressors (Figure 5) as the compressors used in the compression algorithms.*
>
> We thank the reviewer again for his careful and valuable comments. We hope the above response can clarify the reviewer's questions. We are looking forward to the follow-up discussion, and more than happy to address any further comments or questions.

---

### Official Review · Reviewer_oD7w · 2023-07-27

**Soundness:** 3 good
**Presentation:** 3 good
**Contribution:** 3 good
**Rating:** 7
**Confidence:** 2

**Summary:**

The paper explores the conditions of unbiased compression to reduce the communication cost in distributed optimization. Specifically, the paper first presents a theoretical formalization of the total communication cost (TCC) in distributed optimization. With this formulation, the paper proves that unbiased compressor alone cannot necessarily save TCC. Then the paper proves lower bounds on the convergence complexity and shows that independent unbiased compressor provably saves TCC. The paper also improves ADIANA and provides the upper bound of the TCC that can be reduced. Experiments are also provided to support the theoretical findings.

**Strengths:**

Originality:

The paper focuses on how to save TCC, which is a more practical concern in distribution optimization. Though the paper relies on the findings (independence on unbiased compressors) of previous works, it provides new findings (on the upper bound of the TTC that can be reduced) and applicable modifications of existing methods to obtain an match the bound. Overall the paper seems novel to me.

Quality:

The claims in the paper are well supported by theoretical analysis.

Clarity:

The paper is well organized and easy to read. The authors give good introduction and clearly list the questions (Q1, Q2) they need to answer. They also provide detailed intuition on why independent unbiased compressor saves communications.


**Weaknesses:**

Overall the paper looks good to me.  One concern is that the current dataset is relatively small. It would be better if the author can provide experiments on a larger real dataset to better prove the effectiveness of the theoretical findings.



**Questions:**

N/A

---

> ### Author Rebuttal · Authors · 2023-08-02
>
> We thank the reviewer for the positive comments and valuable suggestions. The datasets used in our current manuscript follow the experimental settings in prior works (e.g. [25, 26]) in the literature. According to the reviewer's suggestion, we conducted additional experiments with a larger dataset, CIFAR-10. The new experimental results are shown in Fig. 1 in the pdf attached to our “global response” to all reviewers.
>
> We are looking forward to the follow-up discussion with the reviewer, and more than happy to address any further comments or questions.

---

### Official Review · Reviewer_gfiB · 2023-07-29

**Soundness:** 2 fair
**Presentation:** 2 fair
**Contribution:** 2 fair
**Rating:** 5
**Confidence:** 4

**Summary:**

The paper investigated the overall communication costs for federated learning algorithms. A lower bound on the per-round communication cost is presented, then an analysis of an existing algorithm, ADIANA, is provided to obtain an improved upper bound.

**Strengths:**

The general ideas for the settings and the results are clearly presented. Besides, the presented bounds are clean compared to related works in this field.

**Weaknesses:**

There is an absence of several crucial details, or perhaps typos, which makes it difficult to concretely understand the results. For example, one central concept required in this work is the unbiased compressor, which based on the current manuscript, applies to all real input values. However, if we require it to output a discrete variable with a bounded number of bins, such a compressor may not exist as it is not clear how the unbiased condition could hold for all $x\in\mathbb{R}$.

**Questions:**

There are possible places of improvement in terms of clarity. For example:

1. Generally, it would be better if any concept is defined before it is formally used. For example, in the current manuscript, the definition of “linear spanning” is lost in the texts.

2. Similarly, the assumption of fixed communication load per round could be introduced before the definition of $T_{\epsilon}$ in Sec. 2.4, otherwise, the meaning of minimization in its definition is not clear.

3. For strongly convex results, it might be cleaner to replace $L/\mu$ with $\kappa$.

4. The notation of $U^{int}_{\omega}$ (or $U_\omega$) can be removed (or moved to appendices) if not used.

**Limitations:**

The limitation is appropriately discussed in Section 7.

---

> ### Author Rebuttal · Authors · 2023-08-02
>
> We thank the reviewer for the detailed comments. All questions have been clarified as best as we can. We are glad to address any further comments or questions.
>
> **1. There is an absence of several crucial details, or perhaps typos, which makes it difficult to concretely understand the results.**
>
> The example about unbiased compressors is replied in the next point. We apologize if there are more typos or absence of details that affect the readability. We hope that the reviewer can point them out directly if possible, so that we can give specific responses.
>
> **2. Concept on unbiased compressor.**
>
> The concept of unbiased compressor defined in Assumption 2 is standard in literature. It is consistent with existing concepts on unbiased compressor, such as Eq. (2) in reference [40] on EF21, Definition 2 in reference [26] on CANITA, Definition 1 in reference [25] on ADIANA, Lemma 2 in reference [49] on ECQ-SGD, Assumption 3 in reference [R1] on compressed push-pull, Assumption 4 in reference [13] on NEOLITHIC, Definition 4 and Remark 1 in reference [R2] on DIANA, Lemma 3.1 in reference [3] on QSGD.
>
> Many examples fall into the family of compressor, see Examples 1 and 2 in references [25, 26].
>
> [R1] Z. Song et. al., Compressed Gradient Tracking for Decentralized Optimization Over General Directed Networks, IEEE TSP 2022.
>
> [R2] Samuel Horváth, Dmitry Kovalev, Konstantin Mishchenko, Sebastian Stich, and Peter Richtárik. Stochastic distributed learning with gradient quantization and variance reduction. *arXiv preprint arXiv:1904.05115,* 2019.
>
> **3. It would be better if any concept is defined before it is formally used. The definition of “linear spanning” is lost in the texts.**
>
> We agree. We temporarily put some definitions in the Appendix because of the page limit. For example, the formal definition of “linear spanning” is stated in **Appendix D**. We will move the definition of "linear spanning" in Definition 2 in Appendix D to subsection 2.3 prior to the definition of algorithm class in Definition 1.
>
> **4. The assumption of fixed communication load per round could be introduced before the definition of $T_\epsilon$ in Sec. 2.4, otherwise, the meaning of minimization in its definition is not clear.**
>
> The convergence complexity $T_\epsilon$ is defined as the minimal number of communication rounds needed for guaranteeing an algorithm to find an $\epsilon$-accurate optimum. The notion is customary in literature of communication compression (see, e.g., [20, 25, 32, 25, 26]) and is not explicitly related to the per-round communication load. We chose to state $T_\epsilon$ first as it is widely used and self-explanatory.
> The definitions of per-round communication workload and total communication costs are less investigated in literature. Therefore, we chose to write an entire section to motivate and argue for them.
> As suggested by the reviewer, the comparison in terms of $T_\epsilon$ is more meaningful for algorithms with the same per-round communication cost.
> We thank reviewer gfiB for pointing out the potential misleadingness. We will introduce the matter of the fixed per-round communication cost before $T_\epsilon$  when preparing the camera-ready draft.
>
> Specifically, we will move the following assumption in Line 198~199: *Let each worker equip with a non-adaptive compressor with the same fixed per-round communication cost, i.e., the compressor outputs compressed vectors of the same length (size)* to an additional remark as follows directly after Remark 1 in subsection 2.4.
>
> **Remark 2.** Though the definition of $T_\epsilon$ can be independent of the per-round communication cost, which is specified through the degree of compression $\omega$ (i.e., choice of compressor $C_i$'s), we further assume here that these $C_i$'s equipped by each worker are **non-adaptive compressors with the same fixed per-round communication cost**, i.e., the compressor outputs compressed vectors of the same length (size).
>
> **5. For strongly convex results, it might be cleaner to replace $L/\mu$ with $\kappa$.**
>
> We agree. Thanks for advice.
>
> **6. The notation of $U_\omega^{ind}$ (or$U_\omega$) can be removed (or moved to appendices) if not used.**
>
> These notations are convenient for us to describe our theoretical results (e.g., in line 177, 228, 259, 263) and highlight the difference between ours and [13].
>
>
> We thank the reviewer again for his careful and valuable comments. We hope these response can clarify the reviewer's questions. We are looking forward to the follow-up discussion, and more than happy to address any further comments or questions.

---

> ### Author Response · Authors · 2023-08-14
> **Need more clarifications?**
>
> Dear reviewer gfiB,
>
> We thank you for your valuable comments. We have made detailed responses to address your concerns, but we have not received your replies to our current clarifications yet. We thus kindly ask if our responses have addressed all your concerns. If not, we are more than happy to provide more clarifications.
>
> Best,
>
> The authors of paper 5699

---

> > ### Comment · Reviewer_gfiB · 2023-08-20
> > **Consistency between unbiased compressor and fixed communication**
> >
> > We would like to thank the author for the response. My concern about the unbiased compressor remains, as the communication in this work is measured in bits (as far as I understand, as stated on line 208 or in Figure 1), but the well-known results in [25,26] deliver continuous variables, which require infinitely many bits to make it unbiased. I suppose the authors may be intended to apply an additional quantization step which would introduce some bias, but this is unclear from the manuscripts.
> >
> > Especially when the authors assume fixed and deterministic load per round (instead of variable length), the only class of compression function C that enables decodability has to map the input to a fixed finite set, and any input larger than the maximum in that set can not be mapped to a distribution with unbiased expectation. So for the results to be meaningful, a modification for at least some of the assumptions in the current manuscript is needed. Here are the possibilities.
> >
> > 1. The end-to-end compression function is not unbiased.
> > 2. The communication load can be non-deterministic.
> > 3. The compressor only applies to a subset of the real inputs.
> > 4. There is a random seed shared between all machines and the randomness of C is taken over the shared value.
> > 5. There is an allowed exceptional event with a small probability that the assumptions on the compressors can be violated.
> >
> > It would also be helpful if a concrete example of the unbiased estimator is presented that requires finitely many bits for communication.

---

> > > ### Author Response · Authors · 2023-08-21
> > > **Author response (Part 1/2)**
> > >
> > > We really appreciate the reviewer's deep thought and sharp observation on the concept of unbiased compressors. Below are our responses.
> > >
> > > **1. No finite-bit unbiased compressor over the entire real space.**
> > >
> > > We agree with the reviewer. With finite bits, a compressor cannot estimate an arbitrarily large value in an unbiased manner and thus cannot facilitate the compression for all real numbers/vectors.
> > >
> > > **2. It is a common issue in literature**
> > >
> > > The reviewer's sharp insights indeed expose a common issue existing in a large body of literature. For most literature on unbiased quantization (such as Q-SGD [3] and natural compression [14]), while the quantization schemes therein are defined over the entire real space, they only apply these schemes to values represented by float32 or float64. Apparently, quantizing arbitrarily large (or small) values in the entire real space will result in infinite bits, which contradicts the purpose of saving communication. Thus it makes more sense to quantize values originally represented with float32 or float64 to values with much fewer bits.
> > >
> > > Our work follows this convention. While we define the unbiased compressor over the entire real space, we apply it to input values represented with float32 or float64. Since the input values have already been represented with finite bits, the output values of our defined unbiased compressor are also with finite bits, which is consistent with our setting that "each communication round has fixed and deterministic load".  This convention is also followed by other literature such as [R1], which was accepted by NeurIPS2022, to compare the amount of saved communicated bits in comparison to other baselines.
> > >
> > > [R1] Wang, B., Safaryan, M. and Richtárik, P., 2022. Theoretically better and numerically faster distributed optimization with smoothness-aware quantization techniques. Advances in Neural Information Processing Systems, 35, pp.9841-9852.

---

> > > ### Author Response · Authors · 2023-08-21
> > > **Author response (Part 2/2)**
> > >
> > > **3. A refined definition for unbiased compressor**
> > >
> > > While we are following the convention in literature, we are glad to provide a more precise definition for unbiased compressors stuided in the manuscript to clarify the reviewer's confusion. In fact, an unbiased compressor with finite bits exists if and only if the following conditions hold:
> > >
> > > - **The valid input value must be bounded.** It's clear that using finite bits cannot provide an unbiased estimate for arbitrarily large (or small) values.
> > >
> > > - **The valid non-zero input must be bounded away from zero.** Since the compression uses finite bits, there exists the smallest distance that bounds all possible input values away from 0.
> > >
> > > Given the above conditions, we will modify the classical definition of unbiased compressor as follows:
> > >
> > > **Assumption 2’**
> > >
> > > We assume unbiased compressors $\\{C_i\\}_{i=1}^n$ satisfy
> > >
> > > \begin{equation}
> > > \mathbb{E}[C_i(x)]=x,\quad\mathbb{E}[\\|C_i(x)-x\\|^2]\le\omega\\|x\\|^2,
> > > \end{equation}
> > >
> > > for a constant $\omega\ge0$ and any input $x\in\mathcal{X}\subset\mathbb{R}^d$, where $\mathcal{X}$ satisfies the following condition: **There exists positive constants $0<\epsilon_m<M$ so that it holds for all $ x=(x_1,\cdots,x_d)^\top\in\mathcal{X}$ that $|x_i|\in\\{0\\}\cup[\epsilon_m,M]$, $i=1,\cdots,d$.**
> > >
> > > We believe it is a proper definition because
> > >
> > > - **It is well justified in practice.** The inputs of compressors under machinery computations; therefore, their maximum value and non-zero lower bound are constrained by the computation precision of machines. For instance, when performing computations using float32, the resulting values are bounded above by approximately 3.4e38 and away from zero by at least about 1.18e-38 if non-zero.
> > >
> > > - **It is consistent with unbiased compressors proposed in previous works.** Natural compression [14] is a good unbiased compression scheme, following which we can provide a concrete example of unbiased compressors satisfying Assumption $2^\prime$:
> > >
> > > Letting $\mathcal{C}(0)=0$ and
> > >
> > > \begin{equation}
> > > \mathcal{C}(t)=\left\\{
> > > \begin{array}{2} sign(t)\cdot2^{\lfloor\log_2|t|\rfloor}, &\text{with } p(t),\\\\
> > > sign(t)\cdot2^{\lceil\log_2|t|\rceil}, &\text{with } 1-p(t),
> > > \end{array}
> > > \right.
> > > \end{equation}
> > >
> > > where $t\ne0$ and probability $p(t)=\frac{2^{\lceil\log_2|t|\rceil}-|t|}{2^{\lfloor\log_2|t|\rfloor}}$, it's clear that $d(2+\lceil\log_2(\lceil\log_2(M)\rceil-\lfloor\log_2(\epsilon_m)\rfloor+1)\rceil)$ bits are sufficient for constructing such a compressor that satisfies Assumption $2^\prime$ with $\omega=1/8$.
> > >
> > > - **It does not affect the theoretical results of our work.**
> > >
> > > -- **Upper bound**.  Since the convergence criterion $\epsilon$ is typically much larger than the machine precision $\epsilon_m$ (e.g.,1.18e-38), the distortion incurred by finite bits has little effect on the ideal computations in the real space. In this case,  all inputs of the compressors, which are calculated by machinery computations, naturally fall into the valid input set $\mathcal{X}$ we consider.  As a result, machine precision $\epsilon_m$ does not affect our established upper bound for ADIANA. In fact, it is convection in optimization literature to ignore the effect of machine precision on the convergence rate and complexity.
> > >
> > > -- **Lower bound**. In our paper, we established the lower bound for $\inf_{A\in\mathcal{A}}\sup_{C_i \in \mathcal{U}\_\omega} T(A, \\{C_i\\})$ where $\mathcal{A}$ is the class of algorithms defined in Definition 1,  $\mathcal{U}\_\omega$ is the class of unbiased compressor defined in Assumption 2, and $T(A, \\{C_i\\})$ is the convergence complexity achieved with algorithm $A$ and compressors $C_i$'s. When we consider the new class of unbiased compressor $\mathcal{U}\_\omega^\prime$ specified by Assumption $2^\prime$, it is easy to see that $\mathcal{U}\_\omega \subseteq \mathcal{U}\_\omega^\prime$. Let $T_L$ be the lower bound of $\inf_{A\in\mathcal{A}}\sup_{C_i \in \mathcal{U}\_\omega} T(A, \\{C_i\\})$, it naturally holds that
> > > \begin{equation}
> > > T_L \le\inf_{A\in\mathcal{A}}\sup_{C_i \in \mathcal{U}\_\omega} T(A, \\{C_i\\}) \le \inf_{A\in\mathcal{A}}\sup_{C_i \in \mathcal{U}'\_\omega} T(A, \\{C_i\\})
> > > \end{equation}
> > > In other words, our established lower bound also holds for the new class of unbiased compressors defined by Assumption $2^\prime$.
> > >
> > > Since the upper bound and lower bound are not affected by the newly introduced compressor assumption up to machine precision $\epsilon_m$, they still nearly match each other. In other words, the modified family of unbiased compressors does not affect our theoretical results.
> > >
> > > **Summary**
> > >
> > > We really appreciate the sharp observation and useful suggestions proposed by the reviewer. It is actually a common issue in literature, and it can be practically addressed by considering finite-bit inputs only. We hope this can resolve the reviewer's concerns, and we are looking forward to the follow-up discussions.

---

> > > > ### Comment · Reviewer_gfiB · 2023-08-22
> > > >
> > > > This updated formulation looks good to me, and I'm willing to increase my score from 3 to 5, assuming that the rest of the paper is to be revised to make sure everything is consistent.

---

### Author Rebuttal · Authors · 2023-08-05

We thank all the reviewers for their careful review and valuable feedback. We have addressed each question raised by the reviewers in the separate rebuttals below and are glad to address any further concerns. As several matters have been brought up by multiple reviewers, we present a global response to these concerns below.

**1. Paper organization**

We thank all the reviewers for suggestions on reorganizing some parts of our manuscript, and adding comments or examples for further details. However, the page limit imposes unavoidable sacrifice of the content that we eager to present in the main text. As a result, we have to present details to important concepts and theorems, and refer the rest (e.g., typical examples, standard concepts, customary definitions) to our **Appendix in the Supplementary Material**. We will reorganize our main text by carefully following the advice provided by reviewers (e.g., make modifications as stated in the separate rebuttals) once we have more space allowed when preparing the camera-ready draft.

**2. Concerns related to the lower bounds.**

We find that some concerns of the reviewers are on the understanding of lower bounds. To precisely understand our results, we remark that the lower bound complexity should always be interpreted in the sense of **the limit of best-performing algorithms in convergence (or communication costs) when facing the worst-case instances under pre-specified assumptions of objectives and compressors**. In other words, we aim to investigate the theoretically best-achievable performance under the pre-specified assumptions of objectives and compressors. As a result, the conclusion naturally binds to the setup regularized by pre-specified assumptions.
Once the setup is modified (e.g., the compressors or objective functions are assumed to further enjoy certain properties), the according **worst-case** instance and the best achievable performance also vary.

Consequently, when new assumptions are introduced, some algorithms can surpass the present lower bounds, which does not contradict with our established results since they are with different setups. On the other hand, the communication saving based on the lower bounds is concerned about the worst cases under **the exact settings in our paper**, and any additional assumption on the problem will change the setting and hence may affect our conclusions.

**3. Concern on the use of correlated compressors. Rare works do not use independence.**

We respectfully disagree with the opinion that "rare works do not use independence". Non-IID unbiased compressors can offer practical advantages over IID compressors in many scenarios. Specifically, non-IID random sparsification compressors are more compatible with **All-Reduce** operation, which is a highly effective protocol of distributed gradient aggregation used by default in PyTorch and TensorFlow. However, IID sparsification compressors output compressed gradients with indices of non-zero entries poorly aligned across workers, hindering the efficacy of all-reduce, as illustrated in Fig. 2 in the pdf attached to this "global response". In fact, as noted in [R1, page 7], **gradient compression provides limited benefits if it is not compatible with all-reduce**. To enable unbiased compression to work with all-reduce, compressors across workers must share the same random seed, making them non-IID, as shown in Algorithm 4 of the well-known PowerSGD paper [R2]. In addition, the CSER algorithm also use a shared random seed across compressors [R3, Page 5].

[R1] S. Agarwal et. al., *On the Utility of Gradient Compression in Distributed Training Systems*, arXiv:2103.00543

[R2] T. Vogels et. al,, *PowerSGD: Practical Low-Rank Gradient Compression for Distributed Optimization*, arXiv:1905.13727

[R3] C. Xie et. al., *CSER: Communication-efficient SGD with Error Reset*, arXiv:2007.13221.

**4. More figures**

We provide two additional figures related to our rebuttal in the attached pdf.

**Figure 1.** In Fig.1, we display the results of an additional experiment, where we compare ADIANA with three different compressors and non-compression Nesterov's accelerated algorithm using a more practical dataset, CIFAR-10.

**Figure 2.** In Fig.2, we display the intuition why random sparsification compressors with shared randomness are more compatible with All-Reduce compared with those with independent randomness.

We thank all the reviewers again for the valuable comments. We are looking forward to the follow-up discussion, and more than happy to address any further comments or questions.

---

> ### Author Response · Authors · 2023-08-21
> **A rebuttal summary before the reviewer-author discussion deadline**
>
> Dear Reviewers,
>
> We sincerely appreciate the time and effort you have devoted to reviewing our paper. Your insightful feedback has significantly improved the quality of our work.
>
> We are very delighted that reviewers find that our work "**novel**" and "**provide new findings**" on the lower bound, and establish results that "**match with**" the lower bound, that our claims are "**convincing**" and "**well-supported**" by theoretical analysis, and that our results are "**important and impactful addition**" to the community.
>
> We have taken all questions and concerns seriously, providing comprehensive rebuttals to each reviewer. We are glad that the issues raised by reviewers oD7w, tEWM, u1Ro, MRty, and yR98 have been sufficiently addressed, and we will integrate their feedback into the revision.
>
> We only had the chance to communicate with reviewer gfiB in the last day before the deadline. Our definition of unbiased compressor follows standard convention in literature, i.e., define the compressor over the entire real space but apply it to values represented with float32 or float64. However, to fully resolve reviewer gfiB's concern, we do provide an improved definition of unbiased compressor which is practical, applicable to many compressors established in literature, and does not affect our theoretical results. We hope this can resolve his or her concerns on the concept of unbiased compressor.
>
> While we attempted follow-up with reviewer JbF9, we unfortunately have not received any response. As detailed in our global response point 2, we have provided ample existing work exploiting correlated unbiased compressors. The other concerns are also thoroughly addressed in our response.
>
> We sincerely thank all reviewers for their thoughtful comments and questions. Their feedback has been invaluable for strengthening our work.
>
> Best regards,
>
> Submission 5699 Authors

---

### Decision · Program_Chairs · 2023-09-21

**Decision:**

Accept (poster)

**Comment:**

The reviewers agree on the good contribution of the paper in the area of federated learning. Here are some suggestions for the authors to incorporate: 1) This paper is less about total communication, but rather about the estimation error ($\omega$) of stochastic gradients - and as such, compression issues are completely decoupled, beyond trivially showing that compression lead to some estimation error. This should be made clear; 2) Related to the first issue, the authors seem completely oblivious to compression or communication literature in general, and finally, 3) Even in federated learning, there are *unbiased* compressors and error analysis that this paper does not mention, eg.  Mayekar and Tyagi (AISTATS 2020), Gandikota et al. (AISTATS 2021), and should be included in discussion.